# The Pliocene Model Intercomparison Project Phase 2: Large scale climate features and constraining climate sensitivity.

Alan M. Haywood[1], Julia C. Tindall[1*], Harry J. Dowsett[2], Aisling M. Dolan[1], Kevin M. Foley[2], Stephen J. Hunter[1], Daniel J. Hill[1], Wing-Le Chan[3], Ayako Abe-Ouchi[3], Christian Stepanek[4], Gerrit Lohmann[4], Deepak Chandan[5], W. Richard Peltier[5], Ning Tan[6/7], Camille Contoux[7], Gilles Ramstein[7], Xiangyu Li[8/9], Zhongshi Zhang[8/9/10], Chuncheng Guo[9], Kerim H. Nisancioglu[9], Qiong Zhang[11], Qiang Li[11], Youichi Kamae[12], Mark A. Chandler[13], Linda E. Sohl[13], Bette L. Otto-Bliesner[14], Ran Feng[15], Esther C. Brady[14], Anna S. von der Heydt[16,17], Michiel L. J. Baatsen[17] and Daniel J. Lunt[18].

[1]School of Earth and Environment, University of Leeds, Woodhouse Lane, Leeds, West Yorkshire, LS29JT, UK

[2]Florence Bascom Geoscience Center, U.S. Geological Survey, Reston, VA 20192, USA

[3]Centre for Earth Surface System Dynamics (CESD), Atmosphere and Ocean Research Institute (AORI), University of Tokyo, Japan

[4]Alfred Wegener Institute, Helmholtz Centre for Polar and Marine Research, Bremerhaven, Germany

[5]Department of Physics, University of Toronto, Toronto, Ontario, Canada

[6]Key Laboratory of Cenozoic Geology and Environment, Institute of Geology and Geophysics, Chinese Academy of Sciences, Beijing 100029, China.

[7]Laboratoire des Sciences du Climat et de l'Environnement, LSCE/IPSL, CEA-CNRS-UVSQ, Université Paris-Saclay, F-91191 Gif-sur-Yvette, France

[8]Institute of Atmospheric Physics, Chinese Academy of Sciences, Beijing, China

[9]NORCE Norwegian Research Centre, Bjerknes Centre for Climate Research, Bergen, Norway

[10]Department of Atmospheric Science, School of Environmental Studies, China University of Geosciences, Wuhan, China

[11]Department of Physical Geography and Bolin Centre for Climate Research, Stockholm University, Stockholm, Sweden

[12]Faculty of Life and Environmental Sciences, University of Tsukuba, Tsukuba, Japan

[13]CCSR/GISS, Columbia University, New York, USA

[14]National Center for Atmospheric Research, Boulder, Colorado, USA

[15]Department of Geosciences, College of Liberal Arts and Sciences, University of Connecticut, Connecticut, USA

[16]Centre for Complex Systems Science, Utrecht University, Utrecht, The Netherlands

[17]Institute for Marine and Atmospheric research Utrecht (IMAU), Department of Physics, Utrecht University, Utrecht, The Netherlands.

[18]School of Geographical Sciences, University of Bristol, Bristol, UK.

*Correspondence to*: Julia C. Tindall (earjcti@leeds.ac.uk)

**Abstract.** The Pliocene epoch has great potential to improve our understanding of the long-term climatic and environmental consequences of an atmospheric $CO_2$ concentration near ~400 parts per million by volume. Here we present the large-scale features of Pliocene climate as simulated by a new ensemble of climate models of varying complexity and spatial resolution and based on new reconstructions of boundary conditions (the Pliocene Model Intercomparison Project Phase 2; PlioMIP2). As a global annual average, modelled surface air temperatures increase by between 1.7 and 5.2 °C relative to pre-industrial with a multi-model mean value of 3.2°C. Annual mean total precipitation rates increase by 7% (range: 2%-13%). On average, surface air temperature (SAT) increases by 4.3°C over the land and 2.8°C over the oceans. There is a clear pattern of polar amplification with warming polewards of 60°N and 60°S exceeding the global mean warming by a factor of 2.3. In the Atlantic and Pacific Oceans, meridional temperature gradients are reduced, while tropical zonal gradients remain largely unchanged. There is a statistically significant relationship between a model's climate response associated with a doubling in $CO_2$ (Equilibrium Climate Sensitivity; ECS) and its simulated Pliocene surface temperature response. The mean ensemble Earth system response to doubling of $CO_2$ (including ice sheet feedbacks) is 67% greater than ECS, this is larger than the increase of 47% obtained from the PlioMIP1 ensemble. Proxy-derived estimates of Pliocene sea-surface temperatures are used to assess model estimates of ECS and give a range of ECS between 2.6 and 4.8°C. This result is in general accord with the range in ECS presented by previous IPCC Assessment Reports.

## 1. Introduction

### 1.1 Pliocene climate modelling and the Pliocene Model Intercomparison Project

Efforts to understand climate dynamics during the mid-Piacenzian Warm Period (MP; 3.264 to 3.025 million years ago), previously referred to as the mid-Pliocene Warm Period, have been ongoing for more than 25 years. Beginning with the initial climate modelling studies of Chandler et al. (1994), Sloan et al. (1996) and Haywood et al. (2000), the complexity and number of climate models used to study the MP has since increased substantially (e.g. Haywood and Valdes 2004). This progression culminated in 2008 with the initiation of a co-ordinated international model intercomparison project for the Pliocene (Pliocene Model Intercomparison Project: PlioMIP). PlioMIP Phase 1 (PlioMIP1) proposed a single set of model boundary conditions based on the U. S. Geological Survey PRISM3D data set (Dowsett et al., 2010), and a unified experimental design for atmosphere-only and fully coupled atmosphere-ocean climate models (Haywood et al. 2010, 2011).

PlioMIP1 produced several publications analysing diverse aspects of MP climate. The large-scale temperature and precipitation response of the model ensemble was presented in Haywood et al. (2013a). The global annual mean surface air temperature was found to have increased compared to the pre-industrial, with models showing warming of between 1.8 and

3.6°C. The warming was predicted at all latitudes but showed a clear pattern of polar amplification resulting in a reduced
equator to pole surface temperature gradient. Modelled sea-ice responses were studied by Howell et al. (2016), who demonstrated a significant decline in Artic sea-ice extent, with some models simulating a seasonally sea-ice free Arctic Ocean driving polar amplification of the warming. The reduced meridional temperature gradient influenced atmospheric circulation in a number of ways, such as the poleward shift of the mid-latitude westerly winds (Li et al., 2015). In addition, Corvec and Fletcher (2017) studied the effect of reduced meridional temperature gradients on tropical atmospheric circulation. They demonstrated a weaker tropical circulation during the MP, specifically a weaker Hadley Circulation, and in some climate models also a weaker Walker Circulation, a response akin to model predictions for the future (IPCC, 2013). Tropical cyclones (TC) were analysed by Yan et al. (2016) who demonstrated that average global TC intensity and duration increased during the MP, but this result was sensitive to how much tropical sea surface temperatures (SSTs) increased in each model. Zhang et al. (2013 and 2016) studied the East Asian and West African summer monsoon response in the PlioMIP1 ensemble and found that both were stronger during the MP. Li et al. (2018) reported that the global land monsoon system during the MP simulated in the PlioMIP1 ensemble generally expanded poleward with increased monsoon precipitation over land.

The modelled response in ocean circulation was also studied in PlioMIP1. The Atlantic Meridional Overturning Circulation (AMOC) was analysed by Zhang et al. (2013). No clear pattern of either weakening or strengthening of the AMOC could be determined from the model ensemble, a result at odds with long-standing interpretations of MP meridional SST gradients being a result of enhanced Ocean Heat Transport (OHT: e.g. Dowsett et al., 1992). Hill et al. (2014) analysed the dominant components of MP warming across the PlioMIP1 ensemble using an energy balance analysis. In the tropics increased temperatures were determined to be predominantly a response to direct $CO_2$ forcing, while at high-latitudes changes clear sky albedo became the dominant contributor, with the warming being only partially offset by cooling driven by cloud albedo changes.

The PlioMIP1 ensemble was also used to help constrain Equilibrium Climate Sensitivity (ECS; Hargreaves and Annan 2016). ECS is defined as the global temperature response to a doubling of $CO_2$, once the energy balance has reached equilibrium (this diagnostic is discussed further in section 2.4). Based on the PRISM3 (Pliocene Research, Interpretation and Synoptic Mapping version 3) compilation of MP tropical SSTs, Hargreaves and Annan (2016) estimated that ECS is between 1.9 and 3.7°C. In addition, the PlioMIP1 model ensemble was used to estimate Earth System Sensitivity (ESS). ESS is defined as the temperature change associated with a doubling of $CO_2$ and includes all ECS feedbacks along with long timescale feedbacks such as those involving ice sheets. In PlioMIP1, ESS was estimated to be a factor of 1.47 higher than the ECS (ensemble mean ECS = 3.4°C: ensemble mean ESS = 5.0°C: Haywood et al., 2013a).

*1.2 From PlioMIP1 to PlioMIP2*

The ability of the PlioMIP1 models to reproduce patterns of surface temperature change, reconstructed by marine as well as terrestrial proxies, was investigated via data/model comparison (DMC) in Dowsett et al. (2012; 2013) and Salzmann et al. (2013) respectively. Although the PlioMIP1 ensemble was able to reproduce many of the spatial characteristics of SST and surface air temperature (SAT) warming, the models appeared unable to simulate the magnitude of warming reconstructed at the higher latitudes, in particular in the high North Atlantic (Dowsett et al. 2012, 2013; Haywood et al., 2013a; Salzmann et al. 2013). This problem has also been reported as an outcome of DMC studies for other time periods including the early Eocene (e.g. Lunt et al., 2012). Haywood et al. (2013a, 2013b) discussed the possible contributing factors to the noted discrepancies in DMC, noting three primary causal groupings: uncertainty in model boundary conditions, uncertainty in the interpretation of proxy data and uncertainty in model physics (for example, recent studies have demonstrated that this model-proxy mismatch has been reduced by including explicit aerosol-cloud interactions in the newer generations of models (Sagoo and Storelvmo 2017; Feng et al., 2019)).

These findings substantially influenced the experimental design for the second Phase of PlioMIP (PlioMIP2). Specifically, PlioMIP2 was developed to (a) reduce uncertainty in model boundary conditions and (b) reduce uncertainty in proxy data reconstruction. To accomplish (a), state-of-the-art approaches were adopted to generate an entirely new palaeogeography (compared to PlioMIP1), including accounting for glacial isostatic adjustments and changes in dynamic topography. This led to specific changes, compared to the PlioMIP1 palaeogeography, capable of influencing climate model simulations (Dowsett et al. 2016, Otto-Bliesner et al. 2017). These include the Bering Strait and Canadian Archipelago becoming sub-aerial and modification of the land/sea mask in the Indonesian/Australian region for the emergence of the Sunda and Sahul Shelves. To achieve (b) it was necessary to move away from time-averaged global SST reconstructions, towards the examination of a narrow time slice during the late Pliocene that had almost identical astronomical parameters to the present-day. This made the orbital parameters specified in model experimental design, consistent with the way in which orbital parameters would have influenced the pattern of surface climate and ice sheet configuration preserved in the geological record. Using the astronomical solution of Laskar et al. (2004), Haywood et al. (2013b) identified a suitable interglacial event during the late Pliocene (Marine Isotope Stage KM5c, 3.205Ma). The new PRISM4 (Pliocene Research, Interpretation and Synoptic Mapping version 4) global community-sourced data set of SSTs (Foley and Dowsett, 2019) targets the same interval in order to produce point-based SST data.

Here we briefly present the PlioMIP2 experimental design, details of the climate models included in the ensemble, as well as the boundary conditions used. Following this, we present the large-scale climate features of the PlioMIP2 ensemble focussed solely on an examination of the control MP simulation designated as a CMIP6 simulation (called *midPliocene-eoi400)* and its differences to simulated conditions for the pre-industrial era (PI). We also present key differences between PlioMIP2 and PlioMIP1. PlioMIP2 sensitivity experiments will be presented in subsequent studies. We conclude by presenting the outcomes from a DMC using the PlioMIP2 model ensemble and a newly constructed PRISM4 global compilation of SSTs (Foley and

Dowsett, 2019), and assess the significance of the PlioMIP2 ensemble in understanding Equilibrium Climate Sensitivity (ECS) and Earth System Sensitivity (ESS).

## 2. Methods

### 2.1 Boundary Conditions

All model groups participating in PlioMIP2 were required to use standardised boundary condition data sets for the core *midPliocene-eoi400* experiment (for wider accessibility this experiment will hereafter be referred to as *Plio<sub>Core</sub>*). These were derived from the U.S. Geological Survey PRISM data set, specifically the latest iteration of the reconstruction known as PRISM4 (Dowsett et al. 2016). They include spatially complete gridded data sets at $1° \times 1°$ of latitude/longitude resolution for the distribution of land versus sea, topography, bathymetry, as well as vegetation, soils, lakes and land ice cover. Two versions of the PRISM4 boundary conditions were produced known as standard and enhanced. The standard version of the PRISM4 boundary conditions provides the best possible realisation of Pliocene conditions based around a modern land/sea mask. The enhanced boundary conditions include all reconstructed changes to the land/sea mask and ocean bathymetry. For full details of the PRISM4 reconstruction and methods associated with its development, the reader is referred to Dowsett et al. (2016: this volume).

### 2.2 Experimental Design

The experimental design for *Plio<sub>Core</sub>* and associated PI control experiments (hereafter referred to as *PI<sub>Ctrl)</sub>* was presented in Haywood et al. (2016; this volume), and the reader is referred to this paper for full details of the experimental design. In brief, participating model groups had a choice of which version of the PRISM4 boundary conditions to implement (standard or enhanced). This approach was taken in recognition of the technical complexity associated with the modification of the land/sea mask and ocean bathymetry in some of the very latest climate and earth system models. A choice was also included regarding the treatment of vegetation. Model groups could either prescribe vegetation cover from the PRISM4 dataset (vegetation sourced from Salzmann et al., 2008), or simulate the vegetation using a dynamic global vegetation model. If the latter was chosen, all models were required to be initialised with pre-industrial vegetation and spun-up until an equilibrium vegetation distribution is reached. The concentration of atmospheric $CO_2$ for experiment *Plio<sub>Core</sub>* was set at 400 parts per million by volume (ppmv), a value almost identical to that chosen for the PlioMIP1 experimental design (405 ppmv), and in line with the very latest high-resolution proxy reconstruction of atmospheric $CO_2$ of ~400 ppmv for ~3.2 million years ago using Boron isotopes (De La Vega et al. 2018). However, we acknowledge that there are uncertainties on the KM5c $CO_2$ value, hence the specification of Tier 1 PlioMIP2 experiments (Haywood et al. 2016; this volume), which have $CO_2$ of ~350 ppmv and 450ppmv will be used to investigate $CO_2$ uncertainty at a later date. All other trace gases, orbital parameters and the solar constant were

specified to be consistent with each model's $PI_{Ctrl}$ experiment. The Greenland ice sheet (GIS) was confined to high elevations in the Eastern Greenland Mountains, covering an area approximately 25% of the present-day GIS. The PlioMIP2 Antarctic

ice sheet configuration is the same as PlioMIP1 and has no ice over Western Antarctica. The reconstructed PRISM4 ice sheets have a total volume of $20.1 \times 10^6$ km$^3$, equating to a sea-level increase relative to present day of less than ~24 m (Dowsett et al. 2016; this volume). Integration length was set to be 'as long as possible', or a minimum of 500 simulated years. All modelling groups were requested to fully detail their implementation of PRISM4 boundary conditions, along with the initialisation and spin-up of their experiments in separate dedicated papers that also present some of the key science results

from each model, or family of models (see the separate papers within this special volume: [https://www.clim-past.net/special_issue642.html](https://www.clim-past.net/special_issue642.html)). NetCDF versions of all boundary conditions used for the $Plio_{Core}$ experiment, along with guidance notes for modelling groups, can be found here: [https://geology.er.usgs.gov/egpsc/prism/7.2_pliomip2_data.html](https://geology.er.usgs.gov/egpsc/prism/7.2_pliomip2_data.html).

*2.3 Participating Models*

There are currently 16 climate models that have completed the $Plio_{Core}$ experiment to comprise the PlioMIP2 ensemble. These models were developed at different times and have differing levels of complexity and spatial resolution. A further model HadGEM3 is currently running the $Plio_{Core}$ experiment and results from this model will be compared with the rest of the PlioMIP2 ensemble in a subsequent paper. The current 16 model ensemble is double the size of the coupled atmosphere-ocean ensemble presented in the PlioMIP1 large-scale features publication (Haywood et al. 2013a). Summary details of the included

models, and model physics, along with information regarding the implementation of PRISM4 boundary conditions and each model's ECS can be found in Table 1 and Supplementary Table 1. Each modelling group uploaded the final 100 years of each simulation for analysis. These were then regridded onto a regular $1° \times 1°$ grid using a bilinear interpolation, to enable each model to be analysed in the same way. Means and standard deviations for each model were then calculated across the final 50 years.


*2.4 Equilibrium Climate Sensitivity (ECS) and Earth System Sensitivity (ESS)*

In Section 3.6 we use the $Plio_{Core}$ and $PI_{Ctrl}$ simulations to investigate ECS and ESS. The $Plio_{Core}$ experiments represent a 400 ppmv world that is in quasi-equilibrium with respect to both climate and ice-sheets and hence represents an 'Earth System' response to the 400ppmv $CO_2$ forcing. The 'Earth System' response to a doubling of $CO_2$ (ie 560 ppmv-280 ppmv; ESS) can

then be estimated as follows:

$$ESS = \frac{ln\frac{560}{280}}{ln\frac{400}{280}}(Plio_{Core}[SAT] - PI_{Ctrl}[SAT]) \qquad [1]$$

There will be errors in the estimate of ESS from the above equation. These are due to changes between $Plio_{Core}$ and $PI_{Ctrl}$

which should not be included in estimates of ESS, such as: land-sea mask changes, topographic changes, changes in soil properties and lake changes. However, all these additional changes are likely minimal compared to the ice sheet and GHG changes and are expected to have only a negligible impact on the globally averaged temperature, and therefore the estimate of ESS. For example, Pound et al. (2014) found that the inclusion of Pliocene soils and lake distributions in a climate model had an insignificant effect on global temperature (even though changes regionally could be important).

To assess the relationship across the ensemble between the reported ECS and the modelled ESS, we correlate reported ECS across the ensemble with the associated $Plio_{Core}$ - $PI_{Ctrl}$ temperature anomalies. We do this on a global, zonal mean, and local scale. A strong correlation at a particular location would suggest that MP proxy data at that location could be used to derive a proxy-data constrained estimate of ECS (similar to an "emergent constraint"), while a weak correlation would suggest that proxy-data at that location could not be used in ECS estimates.


**3. Climate Results**

*3.1 Surface air temperature (SAT)*

Fig. 1a shows the global mean surface air temperature (SAT) for each model. The top panel shows the $Plio_{Core}$ and $PI_{Ctrl}$ SATs while the lower panel shows the anomaly between them. In this, and all subsequent figures, the models are ordered by ECS

(see Table 2) such that the model with the highest published ECS (i.e. CESM2; ECS=5.3) is shown on the left, while the model with lowest published ECS (i.e. NorESM1-F; ECS = 2.3) is on the right. Increases in $Plio_{Core}$ global annual mean SATs, compared to each of the contributing models $PI_{Ctrl}$ experiment, range from 1.7 to 5.2 °C (Fig. 1a; Table 2), with an ensemble mean ΔT of 3.2°C. The multi-model median ΔT is 3.0°C, while the 10th and 90$^{th}$ percentiles are 2.1°C and 4.8°C, respectively. Analogous results from individual models of the PlioMIP1 ensemble are shown by the grey horizontal lines on Fig. 1a, and

have a mean warming of 2.7°C. Pliocene warming for individual PlioMIP1 models falls into two distinct anomaly bands that are 1.8 - 2.2°C (CCSM4, GISS-E2-R, IPSLCM5A, MRI2.2) and 3.2 - 3.6°C (COSMOS, HadCM3, MIROC4m, NorESM-L). In general, PlioMIP1 models that were in the lower anomaly band show a larger temperature anomaly in PlioMIP2, while those in the upper anomaly band show a lower temperature anomaly in PlioMIP2. The only exception to this is COSMOS, which is the only PlioMIP2 model to use dynamic vegetation (Table 1), the effect of dynamic vegetation on temperature

anomalies is discussed in Stepanek et al. (2020). PlioMIP2 shows a greater range of responses than PlioMIP1, and PlioMIP2 results are more evenly scattered over the ensemble range. The larger ensemble mean in PlioMIP2 is due to the addition of new and more sensitive models, rather than an increase in the temperature anomaly due to the change in boundary conditions.

PlioMIP2 shows increased SATs over the whole globe (Fig. 1b), with an ensemble average warming of ~2.0°C for the tropical oceans (20°N-20°S), which increases towards the high latitudes (Figs. 1b,c). Multi-model mean SAT warming can exceed 12°C in Baffin Bay and 7°C in the Greenland Sea (Fig. 1b), a result potentially influenced by the closure of the Canadian Archipelago and Bering Strait, as well as by the specified loss of most of the Greenland Ice Sheet (GIS), and the simulated reduction in Northern Hemisphere sea-ice cover (de Nooijer et al., 2020). In the Southern Hemisphere, warming is pronounced in regions of Antarctica that were deglaciated in the MP in both west and east Antarctica (Fig. 1b). Warming in the interior of east Antarctica is limited by the prescribed topography of the MP East Antarctic Ice Sheet (EAIS), which in some places exceeds the topography of the EAIS in the models' $PI_{Ctrl}$ experiments.

In terms of magnitude, the CESM2 model has the greatest apparent sensitivity to imposing MP boundary conditions with a simulated ΔT of 5.2 °C (Fig 1a). This model was published in 2020 and has the highest ECS of all the PlioMIP2 models. This model was not included in PlioMIP1, and its response to Pliocene boundary conditions lies outside the range of all PlioMIP1 models both in global mean and for every latitude band (Figure 1a, 1c). It is also warmer than the PlioMIP2 multi-model mean at nearly all gridboxes (Supplementary Fig. 1). Other particularly sensitive models (EC-Earth3.3, CESM1.2, CCSM4-Utr and CCSM4-UoT; shown as an anomaly from the multi-model mean in Supplementary Fig. 1) are also new to PlioMIP2 and this explains why the simulated ΔT from PlioMIP2 exceeds that from PlioMIP1. The model with the lowest response to PlioMIP2 boundary conditions is the NorESM1-F model, which is also the model with the lowest published ECS. Although there is clearly some correlation between a model's ECS and its $Plio_{Core} – PI_{Ctl}$ temperature anomaly, the relationship is not exact. In particular, the versions of CCSM4 that were run by Utrecht University (CCSM4-Utr) and the University of Toronto (CCSM-UoT) both show a large Pliocene response but have a modest ECS compared to the other models.

Three different versions of CCSM4 contributed to PlioMIP2 (see Table 1): the standard version run at NCAR (hereafter referred to as CCSM) has a simulated ΔT = 2.6°C, while CCSM4-Utr has a simulated ΔT = 4.7°C and CCSM4-UoT has a simulated ΔT = 3.8°C. A notable difference between these simulations is the response in the 60°S - 90°S band where the mean warming in the CCSM4-Utr simulation is 4 °C higher than in the CCSM4-UoT simulation and 6.6 °C higher than in the CCSM4 simulation (Fig. 1c; Supplementary Fig. 1). Supplementary Table 1 shows that even though the CCSM4 models differ in their response they all appear to be close to equilibrium. In addition, they are all reported to have similar ECS (Table 1) and they all have the same physics apart from changes to the standard ocean model in the CCSM4-UoT simulations and the $Plio_{Core}$ CCSM4-Utr simulation. These changes (discussed by Chandan & Peltier, 2017, this volume) are: 1. the vertical profile of background diapycnal mixing has been fixed to a hyperbolic tangent form, and 2. tidal mixing as well as dense water overflow parameterization schemes have been turned off. Although the exact cause of the differences in ΔT between the CCSM4 models remains unclear, the changes in the ocean parameterisations and differences in initialization may contribute to the ΔT differences, in particular the changes in ocean mixing between different versions of the model (Fedorov et al., 2010).

Analysis of the standard deviation of the model ensemble (Fig. 1d) indicates that models are generally consistent in terms of
the magnitude of temperature response in the tropics, especially over the oceans. However, they can differ markedly in the
higher latitudes, where the inter-model standard deviation reaches more than 4.5°C.

To evaluate whether the multi-model mean $Plio_{Core} - PI_{Ctrl}$ anomaly at a gridbox is "robust" we follow the methodology of
Mba et al. (2018) and Nikulin et al. (2018). The anomaly is said to be "robust" if two conditions are fulfilled: (1) at least 80%
models agree on the sign of the anomaly, and (2) the signal to noise ratio (i.e. the ratio of the size of the mean anomaly to the
inter-model standard deviation [Fig. 1b / Fig. 1d]) is greater than or equal to one. Regions where the SAT anomaly is
considered robust according to these criteria are hatched in Fig. 2. It is seen that for SAT the $Plio_{Core} - PI_{Ctrl}$ anomaly is
considered robust across the ensemble over nearly all the globe.

*3.2 Seasonal cycle of surface air temperature, land/sea temperature contrasts and polar amplification*

The Northern Hemisphere (NH) averaged SAT anomaly over the seasonal cycle is presented in Fig. 3a. Overall, the ensemble
mean anomaly (black dashed line) is fairly constant throughout the year, however, models within the ensemble have very
different characteristics in terms of the monthly and seasonal distribution of the warming. Some members of the ensemble
have a relatively flat seasonal cycle in ΔSAT (e.g. NorESM-L, NorESM1-F, COSMOS), however others show a very strong
seasonal cycle. The models that show a very strong seasonal cycle do not agree on the timing of the peak warming. For
example, EC-Earth3.3 has the peak warming in October, CESM2 has peak warming in July and MRI2.3 has peak warming in
Jan/Feb, The lack of consistency in the seasonal signal of warming has interesting implications in terms of whether PlioMIP2
outputs could be used to examine the potential for seasonal bias in proxy data sets. To do this meaningfully would require
clear consistency in model seasonal responses, which is absent in the PlioMIP2 ensemble. The grey shaded area in Fig 3a
shows the range of NH temperature response in PlioMIP1, with the PlioMIP1 ensemble average shown by the black dotted
line. Although the ensemble average from PlioMIP2 and PlioMIP1 both show a relatively flat seasonal cycle, the range of
responses is very different between the two ensembles. PlioMIP1 predicted a large range of temperature responses in the NH
winter, which reduced in the summer. In PlioMIP2, however, the summer range is amplified compared to the winter. Indeed
7 of the 16 PlioMIP2 models show a NH summer temperature anomaly that is noticeably above that seen in any of the PlioMIP1
simulations. Some of these models (CESM2, EC-Earth3.3, CCSM4-Utr, CCSM4-UoT and CESM1.2) did not contribute to
PlioMIP1, showing that which models are included in an ensemble can strongly affect the ensemble response. However other
models (MIROC4m and HadCM3) that show an enhanced summer response in PlioMIP2 were also included in PlioMIP1,
showing that there is also an impact of the change in boundary conditions on seasonal temperature. None of the PlioMIP2
models replicate the lowest warming seen in DJF in the PlioMIP1 ensemble, this lowest value was derived from the GISS-E2-
R model in PlioMIP1 which did not contribute to PlioMIP2.

The ensemble results for land/sea temperature contrasts clearly indicate a greater warming over land than over the oceans (Fig. 3b). This result also holds when only the land/sea temperature contrast in the tropics is considered. The land amplification factor is similar in PlioMIP2 and PlioMIP1, and models in both ensembles cluster near a land amplification factor of ~1.5. There is also no relationship between a model's climate sensitivity and the land amplification factor. The multi-model median (10th percentile / 90th percentile) warming over the land and ocean is 4.5°C (2.6°C / 6.1 °C) and 2.5°C (1.9°C / 4.4 °C) respectively.

The extratropical NH (45°N-90°N) warms more than the extratropical Southern Hemisphere (SH) (45°S-90°S) in 5 of the 8 models (62%) from PlioMIP1 and in 11 of the 16 models (69%) from PlioMIP2 (Fig. 3c). This shows that neither the change in boundary conditions nor the addition of newer models to PlioMIP2 affects the ensemble proportion of enhanced NH warming. Neither does the published ECS have any obvious impact on whether the warming is concentrated in the NH or the SH. The models that indicate greater SH versus NH warming (CCSM4-Utr, GISS2.1G, NorESM-L), are among those that have weaker differences between land and ocean warming (Fig 3b).

Polar amplification (PA) can be defined as the ratio of polar warming (poleward of 60° in each hemisphere) to global mean warming (Smith et al 2019). The PA for each model for the NH and the SH is shown in Fig. 3d. All models show PA > 1 for both hemispheres, although whether there is more PA in the NH or SH is a model dependent feature. The ensemble mean (median) PA is 2.3 (2.2) in both the NH and the SH, suggesting that across the ensemble PA is hemispherically symmetrical. This result is very similar to PlioMIP1 (not shown), which suggests that the enhanced warming in the PlioMIP2 ensemble does not affect the PA. For PlioMIP2, the NH median PA is 2.2, with the 10th and 90th percentiles at 1.9 and 2.8 respectively, while in the SH the median PA is 2.2, with the 10th and 90th percentiles at 1.8 and 3.1 respectively. Polar amplification is lower over the land than the ocean (Supplementary Figure 2) in both hemispheres. The NH mean (10th / 50th / 90th percentiles) PA is 1.6 (1.4 / 1.6 / 1.9) and 2.7 (2.4 / 2.7 / 3.3) over the land and ocean respectively, while the SH mean (10th / 50th / 90th percentiles) PA is 0.9 (0.5 / 0.8 / 1.5) and 1.9 (1.1 / 1.9 / 2.5) over the land and ocean respectively. Note that in the SH total PA is higher than both land and ocean PA because of the change in the area of land between the $Plio_{Core}$ and $PI_{Cntl}$ experiments. There appears to be a weak relationship between the PA factor and a model's ECS. Those models which have a lower published ECS (those to the right of Fig. 3d) have a tendency towards higher PA. This is not because these models have excess warming at high latitudes, rather these models have less tropical warming than other models.

*3.3 Meridional/zonal SST gradients in the Pacific and Atlantic*

There has been great interest in the reconstruction of Pliocene SST gradients in the Atlantic and Pacific to provide first order assessments of Pliocene climate change, and to assess possible mechanisms of Pliocene temperature enhancement and ocean/atmospheric dynamic responses (Rind and Chandler, 1991). For example, the meridional gradient in the Atlantic has been discussed in terms of the potential for enhanced Ocean Heat Transport in the Pliocene (e.g. Dowsett et al., 1992). In

addition, the zonal SST gradient across the tropical Pacific has been used to examine the potential for change in Walker Circulation and, through this, ENSO dynamics and teleconnection patterns during the Pliocene (Fedorov et al., 2013; Burls and Fedorov, 2014; Tierney et al., 2019).

The multi-model mean meridional profile of zonal mean SSTs in the Atlantic Ocean is shown in Fig. 4a. In the tropics and sub-tropics, the SST increase between the $Plio_{Core}$ and $PI_{Ctrl}$ experiments is 1.5-2.5°C. This difference increases to ~5.0°C in the NH at ~55°N, with an inter-model range of 2°C - 11°C. The Pliocene and Pre-industrial meridional SST profile in the Pacific (Fig. 4b) is similar to that of the Atlantic, but with little indication from the multi-model mean for a high latitude enhancement in meridional temperature. However, a large range in the ensemble response is noted, and the importance of an

adjustment of the vertical mixing parameterization towards simulation of a reduced Pliocene meridional gradient has been recently shown (Lohmann et al., in review).

     In the tropical Atlantic (20°N -20°S) the multi-model mean zonal mean SST for the Pliocene increases by ~1.9°C (ensemble range from 0.8°C to 3-4°C), with a flat zonal temperature gradient across the tropical Atlantic (Fig. 4c). In the tropical Pacific both Pliocene and pre-industrial ensembles clearly show the signature of both a western Pacific Warm Pool, and the relatively

cool waters in the eastern Pacific that are associated with upwelling (Fig. 4d). As such, a clear east-west temperature gradient is evident in the Pliocene tropical Pacific in the PlioMIP2 ensemble (similar to PlioMIP1) and is not consistent with a permanent El-Niño (see Supplementary Fig. 3). The PlioMIP2 ensemble supports a recent proxy-derived reconstruction for the Pacific that found Pliocene ocean temperatures increased in both the eastern and western Tropical Pacific (Tierney et al., 2019).

Using the methodology of Mba et al. (2018) and Nikulin et al. (2018), the signal of SST change seen in the multi-model mean is robust over nearly all ocean grid cells (Supplementary Fig. 4). Supplementary Fig. 3 shows the difference between the Pliocene ΔSST for each model in the PlioMIP2 ensemble and the Pliocene ΔSST of the multi-model mean. This illustrates that despite the climate anomaly being larger than the inter-model standard deviation there are still many regions (e.g. Southern Ocean, North Atlantic Ocean, Arctic Ocean) where there is a notable inter-model spread of the magnitude of the Pliocene SST

anomalies.

*3.4 Total precipitation rate*

     Simulated increases in $Plio_{Core}$ global annual mean precipitation rates, compared to each contributing model's $PI_{Ctrl}$ experiment, (hereafter referred to as ΔPrecip) ranges from 0.07 to 0.37 mm/day (Fig. 5a), which is notably larger than the

340 PlioMIP1 range of 0.09-0.18 mm/day (shown as horizontal grey lines on Fig. 5a). The PlioMIP2 ensemble mean ΔPrecip is 0.19 mm/day. The increase in the globally averaged precipitation anomaly in PlioMIP2 is due to the addition of new models to the ensemble, which have high ECS and are also more sensitive to the PlioMIP2 boundary conditions. Models that were included in PlioMIP1 (COSMOS, IPSLCM5A, MIROC4m, HadCM3, CCSM4, NorESM-L and MRI2.3) show PlioMIP2

precipitation anomalies that are similar to PlioMIP1 results. The spatial pattern (Fig. 5b) shows enhanced precipitation over
high latitudes and reduced precipitation over parts of the subtropics. The largest ΔPrecip is found in the tropics, in regions of
the world that are dominated by the monsoons (West Africa, India, East Asia). The enhancement in precipitation over North
Africa is consistent with previous Pliocene modelling results that have demonstrated a weakening in Hadley Circulation linked
to reduced pole to equator temperature gradient (e.g. Corvec and Fletcher 2017). Greenland shows increased $Plio_{Core}$
precipitation in regions that have become deglaciated and are therefore substantially warmer. Latitudes associated with the
westerly wind belts also show enhanced $Plio_{Core}$ precipitation, with an indication of a poleward shift in higher latitude
precipitation. This result is consistent with findings from PlioMIP1 (Li et al. 2015). Other, more locally defined ΔPrecip
appears closely linked to localised variations in Pliocene topography and land/sea mask changes, for example, the Sahul and
Sunda Shelf that become subaerial in the $Plio_{Core}$ experiment. In general, the models that display the largest SAT sensitivity
to the prescription of Pliocene boundary conditions also display the largest ΔPrecip (CESM2, CCSM4-Utr, EC-Earth3.3). This
is consistent with a warmer atmosphere leading to a greater moisture carrying capacity and therefore greater evaporation and
precipitation. The model showing the least sensitivity in terms of precipitation response is GISS2.1G.

Analysis of the standard deviation within the ensemble demonstrates that, in contrast to SAT, models are most consistent
regarding ΔPrecip in the extratropics (Fig. 5c). This is similar to the findings of PlioMIP1 (Haywood et al., 2013a) and is
likely because more precipitation falls in the tropics rather than extratropics, and therefore the inter-model differences are
larger in the tropics. The methodology of Mba et al. (2018) and Nikulin et al. (2018) (described in section 3.1), was used to
determine the robustness of ΔPrecip (Fig. 5d). Unlike the temperature signal, which was robust throughout most of the globe,
there are large regions in the tropics and subtropics where the ensemble precipitation signal is uncertain. Changes in
precipitation rates in the subtropics have some inter-model coherence in many places because at least 80% of models agree
on the sign of change. However, most of these predicted changes are not robust because the magnitude of change is not large
compared to the standard deviation seen in the ensemble (Fig. 5c). This is consistent with results from CMIP5, which show
predicted precipitation changes have low confidence particularly in the low and medium emissions scenarios (IPCC, 2013).
The signal of precipitation change is determined to be robust in the high latitudes and in the mid-latitudes in regions influenced
by the westerlies. This is also the case in regions influenced by the West African, Indian and East Asian Summer Monsoons
(Fig. 5d). Supplementary Fig. 5 shows the difference between each model's ΔPrecip and the multi-model mean ΔPrecip
(shown in Fig. 5b), highlighting that there is uncertainty in the ensemble with respect to the regional patterns of precipitation
change.

*3.5 Seasonal cycle of total precipitation and land/sea precipitation contrasts*

Figure 6a shows the seasonal cycle of the precipitation anomaly averaged over the Northern Hemisphere. As was the case for
SAT, the monthly and seasonal distribution of precipitation anomalies are highly model dependent, although the ensemble

average shows a clear NH late spring to autumn *Plio*$_{Core}$ enhancement in precipitation (Fig. 6a). This is most strongly evident in the models CESM2, EC-Earth3.3 and CCSM4-Utr, however it is also evident in other models. Some models show a different seasonal cycle to the annual mean, for example the GISS2.1G model simulates the NH late spring to autumn ΔPrecip being supressed compared to the rest of the year, and HadCM3 which has a bimodal distribution. An increase in NH summer precipitation is consistent with a general trend of West African, Indian and East Asian Summer monsoon enhancement, and this will be discussed in detail in a forthcoming PlioMIP2 paper. In PlioMIP1 (ensemble average - dotted black line and model range – shaded grey area in Fig. 6a) the seasonal cycle in precipitation was much more muted. PlioMIP1 results in the boreal winter are similar to PlioMIP2, however the mean precipitation anomaly in PlioMIP2 between June and November is 40% larger than PlioMIP1. This increase is mainly due to the inclusion of new and more sensitive models into PlioMIP2 (e.g. CESM2 and EC-Earth3.3). However, some models with enhanced summer precipitation (e.g. COSMOS) contributed to both PlioMIP1 and PlioMIP2 suggesting a role of boundary condition changes in enhancing the NH boreal summer precipitation. It is noted, however, that not all the new models in PlioMIP2 show enhanced summer precipitation relative to PlioMIP1. The GISS2.1G model, which was new to PlioMIP2, shows the most muted summer precipitation response in the NH in all PlioMIP2 and PlioMIP1 models. This means that the range of summer/autumn NH precipitation responses as shown by the ensemble increases significantly in PlioMIP2. For example, PlioMIP1 showed a NH precipitation response in October to be 0.13-0.42 mm/day while in PlioMIP2 this has increased to 0.05-0.70 mm/day.

In terms of the land/sea ΔPrecip contrast the PlioMIP2 ensemble divides into two groups (Fig. 6b). One in which a clear pattern of precipitation anomaly enhancement over land compared to the oceans is seen (EC-Earth3.3, MIROC4m, HadCM3, CCSM4, CCSM4-Utr, CCSM4-UoT, NorESM-L and NorESM-F) and the other where there is either a small or no enhancement in the land versus oceans (CESM2, IPSLCM6A, COSMOS, CESM1.2, IPSLCM5A, IPSLCM5A2, GISS2.1G and MRI2.3). Models which show the greatest precipitation enhancement over the land are generally those with a lower published ECS (those to the right of Fig 6b), which have a small precipitation response over the oceans but have a land precipitation anomaly similar to other models. Models with higher ECS (e.g. CESM2) show a similar precipitation anomaly over the land and ocean. Grey horizontal lines on Fig. 6b shows the land/sea ΔPrecip amplification for the PlioMIP1 models. None of the PlioMIP1 models have a ΔPrecip amplification factor > 2, however half of the PlioMIP2 models do. Further, four models which contributed to both PlioMIP1 and PlioMIP2 (MIROC4m, HadCM3, CCSM4, NorESM-L) show a much greater land amplification in PlioMIP2, showing that the change in boundary conditions strongly affects this diagnostic.

*3.6 Climate and Earth System Sensitivity*

This section will consider the relationship between ECS and ESS across the ensemble. Table 2 shows the ECS for each model (referenced in Table 1) and the ESS estimated from the *Plio*$_{Core}$ − *PI*$_{Ctrl}$ temperature anomaly (equation 1). Due to the prescribed changes to ice sheets and vegetation, the *Plio*$_{Core}$ simulation is representing a state in which the associated feedbacks

are in equilibrium. The mean ESS / ECS ratio is 1.67, suggesting that the ESS based on the ensemble is 67% larger than the ECS, however the range is large with the GISS2.1G model suggesting that the ESS / ECS ratio is 1.22 while the CCSM4_Utr model suggests that the ESS / ECS ratio is 2.85

The first analysis of how ECS relates to ESS will consider the correlation between ECS and the globally averaged $Plio_{Core}$ - $PI_{Ctrl}$ temperature anomaly. This is seen in Fig. 7a, and each cross represent the results from a different model in the PlioMIP2 ensemble. There is a significant relationship between ECS and the $Plio_{Core}$ - $PI_{Ctrl}$ temperature anomaly at the 95% confidence level ($p=0.01$, $R^2=0.35$) with the line of best fit: ECS = 2.3 + (0.44 × ($Plio_{Core}(SAT)$ - $PI_{Ctrl}(SAT)$)).

Next, we investigate whether there is a correlation across the ensemble between ECS and the $Plio_{Core} - PI_{Ctrl}$ SAT anomaly on spatial scales. In the analysis that follows we will simply assess whether such a correlation exists and if so, how strong it is, by looking at *p-values* and *R-squared* values, calculated from the models in the ensemble. Fig. 7b shows the relationship (*p-value* – blue, *R-squared* - red) across the ensemble between modelled ECS and the modelled zonal mean $Plio_{Core}$ - $PI_{Ctrl}$ SAT anomaly. We find a significant relationship ($p < 0.05$) between ECS and the zonal mean Pliocene temperature anomaly throughout most of the tropics. This relationship becomes significant at the 99% confidence level ($p < 0.01$) between 38°N and 27°S, where a high proportion of the inter-model variability in global ECS can be related to the inter-model variability in the Pliocene SAT anomaly at an individual latitude, reaching a maximum of 65% at ~15°N.

Next the relationship between global ECS and the local $Plio_{Core} - PI_{Ctrl}$ SAT anomaly is assessed. In Fig. 7c colours show the *R-squared* correlation across the ensemble between modelled global ECS and modelled local $Plio_{Core} - PI_{Ctrl}$ SAT anomaly. The regions where the relationship between the two is significant at the 95% confidence level is hatched. The relationship between ECS and the local $Plio_{Core} - PI_{Ctrl}$ SAT anomaly is significant over most of the tropics, and over some mid and high latitude regions including Greenland and parts of Antarctica. In many cases, the tropical oceans show a temperature anomaly more strongly related to ECS than the land, although this is not always the case.

## 4. Data/Model Comparison

Haywood et al (2013a, b) proposed that the proxy data/climate model comparison in PlioMIP1 could include discrepancies owing to the comparison between time averaged PRISM3D SST and SAT data, and climate model representations of a single time slice. In order to improve the integrity of the data/model comparisons in PlioMIP2, Foley and Dowsett (2019) synthesised alkenone SST data that can be confidently attributed to the MIS KM5c time slice that experiment $Plio_{Core}$ is designed to represent. Foley and Dowsett (2019) provide two different SST data sets. One data set includes all SST data for an interval of 10,000 years around the time slice (5,000 years to either side of the peak of MIS KM5c) and the other covers 30,000 years (up to 15,000 years to either side of the peak; this latter dataset will hereafter be referred to as F&D19_30). Age models used in the compilation are those originally released with the data sets, but later modifications of age models or the integration of additional data could result in mean SST values different from those reported in F&D19_30. All SST estimates are calibrated

using Müller et al. (1998). Prescott et al. (2014) demonstrated that due to the specific nature of orbital forcing 20,000 years before and after the peak of MIS KM5c, age and site correlation uncertainty within that interval would be unlikely to introduce significant errors into SST-based DMC. Given this, and in order to maximise the number of ocean sites where SST can be derived, we carry out a point-based SST data/model comparison using the F&D19_30 data set.

We compare the multi-model mean SST anomaly to a proxy SST anomaly created by differencing the F&D19_30 data set
from observed pre-industrial SSTs derived for years 1870-1899 of the NOAA ERSST version 5 data set (Huang et al., 2017; Fig. 8a and Fig. 8b). Fig. 8c shows the proxy data ΔSST minus the multi-model mean ΔSST. Using the multi-model mean results, 17 of the 37 sites show a difference in model/data ΔSST of no greater than +/- 1°C (Fig. 8c). These are located mostly in the tropics, but also include sites in the North Atlantic, along the coastal regions of California and New Zealand and in the North Pacific. In terms of discrepancies, the clearest and most consistent signal comes from the Benguela upwelling system
(off the south west coast of Africa) where the multi-model mean does not predict the scale of warming seen in 3 of the 4 proxy reconstructions. The multi-model mean is insufficiently sensitive in the two Mediterranean Sea sites, along the east coast of North America (Yorktown Formation), and at one location west of Svalbard close to the sea-ice margin. The multi-model mean predicts too great a warming at one location off the Florida and Norwegian coasts. No discernible spatial pattern or structure is seen (outside of the Benguela and Mediterranean regions) for sites where the ensemble under or overestimates the
magnitude of SST change.

Comparing model predicted and proxy based absolute SST estimate for the MIS KM5c time slice (Fig. 8d) yields a similar outcome to the comparison of SST anomalies (Fig. 8c). However, the Benguela region shows greater model-data agreement when considering absolute SSTs than when considering anomalies. Furthermore, a somewhat clearer picture emerges of the model ensemble not producing SSTs that are warm enough in the higher latitudes of the North Atlantic and especially Nordic
Sea. Although this appears site dependant as the ensemble overestimates absolute SSTs near Scandinavia.

The proxy data ΔSST minus the mean ΔSST for individual models is shown in Supplementary Fig. 6. In regions where there was a strong discrepancy between the proxy data ΔSST and the multi-model mean ΔSST none of the individual models show good model-data agreement. The EC-Earth3.3 model shows an improved agreement with the data in the Mediterranean, the Benguela upwelling system, the site along the East Coast of North America and the site to the West of Svalbard. However,
this improved model-data agreement is at the expense of reduced model-data agreement elsewhere: many of the low and mid-latitude sites, which had good model-data agreement for the multi-model mean have reduced model-data agreement in EC-Earth3.3. Other models, which showed large warming in PlioMIP2 (i.e. CESM2 and CCSM4-Utr) also show a larger ΔSST than the data for some of the tropical and mid-latitude sites which were in good agreement with the multi-model mean. Models that were less sensitive to Pliocene boundary conditions (i.e. GISS2.1G and NorESM-L) do not predict the amount of warming
seen in the data for some of the North Atlantic sites and the multi-model means performs better. Table 3 shows statistics for the data-model comparison for both individual models and the multi-model mean. The root mean square error (RMSE) between the model and the data is 3.72 for the multi-model mean, but is lower in some individual models (namely CESM2,

IPSLCM6A, EC-Earth3.3, CESM1.2, CCSM4-UoT). In general, those models that have a lower model-data RMSE are those which have higher ECS and a higher $Plio_{Core} - PI_{Ctrl}$ warming, while less sensitive models have a higher model-data RMSE.

The average difference between the data and model across all the data points shows a similar pattern. The proxy data is on average 1.5°C warmer than the multi-model mean. However, some individual models have a much smaller average model-data discrepancy (e.g. CESM2 = -0.18°C). The models with a lower model-data discrepancy are those which also have a lower model-data RMSE and have higher than average $Plio_{Core} - PI_{Ctl}$ warming.

This initial analysis suggests that the most sensitive models agree better with the proxy data than the less sensitive models.
However further analysis does not fully support this result. If we consider how many of the 37 sites have 'good' model-data agreement a different picture emerges. Table 3 shows how many sites have model ΔSST within 2°C, 1°C and 0.5°C of the data ΔSST. Using these diagnostics, the MMM performs better than any of the 16 individual models. Those models which have the lowest RMSE and the best average model-data agreement are not those models which have largest number of sites where model and data agree. For example, CESM2 and EC-Earth3.3 have a particularly low number of sites with good model-
data agreement. The models with the highest number of sites with model-data agreements (e.g. ISPLCM6A and CCSM4-UoT, MIROC4m and CESM1.2) show a $Plio_{Core} - PI_{Ctl}$ warming that is closer to the MMM. The fact that the MMM has more sites with 'good' model-data agreement than any individual model, highlights the benefit of performing a large multi-model ensemble as we have done for PlioMIP2. It allows inherent biases within individual models to cancel out and likely provides a more accurate way of estimating climate anomalies than can be done with a single model.

Models show a strong relationship between SST anomalies and global mean SAT anomalies (Supplementary Figure 7a; SATanom = (1.18 × SSTA) + 0.66, Rsq=0.97); and also a strong correlation between SST averaged over 60°N-60°S and global mean SAT anomalies (Supplementary Figure 7b; ΔSAT = (1.16 × ΔSST) + 0.74, with Rsq=0.97). This strong correlation suggests that proxy-based SST anomaly estimates can be used to infer global mean SAT anomalies, provided that enough SST proxy data is available to reliably estimate SST anomalies. The multi-model median ratio of ΔSAT / ΔSST is 1.4, while the
multi-model median ratio of ΔSST to ΔSST (60°N-60°S) is 1.5.

## 5. Discussion

### 5.1 Large-scale features of a warmer climate (palaeo vs future, older vs younger models)

The range in the global annual mean ΔSAT shown by the PlioMIP2 ensemble (from 1.7 to 5.2°C) is akin to the best estimate
(and uncertainty bounds) of predicted global temperature change by 2100CE using the RCP4.5 to 8.5 scenarios (RCP4.5 = 1.8 ± 0.5 °C and RCP8.5 = 3.7 ± 0.7 °C IPCC, 2013; Table 12.2). Comparing the degree of Pliocene temperature change to predicted changes at 2300 CE, the multi-model mean SAT change is between RCP4.5 (2.5 +/- 0.6 °C) and RCP6.0 (4.2 +/- 1.0 °C).

Studies have suggested that the Arctic temperature response to a doubling of atmospheric $CO_2$ concentration may be 1-3 times
that of the global annual mean temperature response (Hind et al., 2016). All 16 models within the PlioMIP2 ensemble simulate
a polar amplification factor (PA; averaged over the NH and SH) between 2 and 3 (meaning that the high latitude temperature
increase is 2-3 times the global mean temperature increase) however, 2 models (GISS2.1G and NorESM-L) show PA > 3 in
the SH. An important caveat to note in the comparison between Pliocene and future predicted polar amplification factors is
the major changes in the size of the ice sheets, which in terms of area-of-ice difference affect the SH far more than the NH.

Both model simulations and observations (Byrne and O'Gorman, 2013) show that as temperatures rise, the land warms more
than the oceans. This is due to differential lapse rates linked to moisture availability on land. From a theoretical standpoint
the difference in land/sea warming is expected to be monotonic with increases in temperature. However, in reality the rise is
non-monotonic and is regulated by latitudinal and regional variations in the availability of soil moisture that influences lapse
rates (Byrne and O'Gorman, 2013). This is evident in the PlioMIP2 ensemble with land/sea amplification of warming noted
more strongly in the global mean than in the tropics where precipitation is most abundant (Fig. 3b). For perturbations to the
pre-industrial, modelling and observational studies have shown that land warms 30 to 70% more than the oceans (Lambert and
Webb, 2011). The PlioMIP2 ensemble broadly supports this conclusion and previous work. It also supports studies that have
indicated that the land/sea warming contrast is not dependent upon whether we are considering a transient (RCP-like) or an
equilibrium-type climate change scenario (e.g. Lambert & Webb, 2011).

In predictions of future climate change, a consistent result from models is that the warming signal is amplified in the Northern
compared to Southern Hemisphere in the extratropics. There have been several studies which have proposed mechanisms to
explain this, including heat uptake by the Southern Ocean (Stouffer et al., 1989) as well as ocean heat transport mechanisms
(Russell et al., 2006). Within the PlioMIP2 ensemble, 11 out of 16 models show a larger temperature change in the NH
extratropics than the SH extratropics (Fig. 3c). This can in part be explained by the area of land in the NH being larger than
in the SH and the already discussed amplification of warming over the land versus the oceans. However, the degree of
difference is highly model dependent and not as large as has been reported for simulation of future climate change by the IPCC
(IPCC, 2013). This may be linked to the intrinsic difference in response between a RCP-like transient and equilibrium climate
experiment, and in the Pliocene substantially reduced ice sheets on Antarctica, which are not specified in future climate change
simulations. Hence, the noted hemispheric difference in warming for the future may simply be a transient feature that would
not be sustained as the ice sheets on Antarctica responded to the warming over centennial to millennial timescales.

The 3.2°C increase in multi-model mean temperature is associated with a 7% increase in global annual mean precipitation.
According to the Clausius-Clapeyron equation, the water holding capacity of the atmosphere increases by about 7% for each
1°C of temperature increase. The increase in precipitation is therefore less than would be expected if it were assumed that all
aspects of the hydrological cycle remained the same as pre-industrial. This is in line with model simulations of future climate
change linked to greater temperatures enhancing evaporation from the surface and the atmosphere having a greater moisture
carrying capacity, but sluggish moist convection (Held and Soden, 2006).

A particularly robust feature of precipitation change across the ensemble is over the modern Sahara Desert and over the Asian monsoon region (Figure 5d). These regions also experience enhanced precipitation under the RCP8.5 scenario for 2100 (IPCC, 2013; Figure SPM.7). However, in other tropical and subtropical regions the $Plio_{Core}$ model response is small compared to the
pre-industrial inter-model standard deviation.

Corvec and Fletcher (2017) showed that in PlioMIP1 studies the tropical overturning circulations in the mPWP were weaker than pre-industrial simulations, while Sun et al. (2013) showed that both Hadley Cells expanded polewards, a result consistent with (but weaker than) the RCP4.5 scenario. These changes in circulation are consistent with the expansion of the subtropical highs and the corresponding reduction in subtropical oceanic precipitation seen in Figure 5 for the $Plio_{Core}$ ensemble and in
IPCC, (2013; Figure SPM.7) for RCP scenarios at year 2100.

Although there are many similarities in tropical atmospheric circulation response between Pliocene experiments and the RCP future climate change experiments, there are specific differences mainly relating to a) the ice sheet changes and their effects on the equator-to-pole temperature gradient during the Pliocene vis-à-vis the future, and b) the fixed vs transient GHG changes. Nonetheless the similarities between the general features of the Pliocene experiments and future experiments continues to
support the use of the Pliocene as one of the best geological analogues for the near future (Burke et al. 2018), despite the different boundary conditions.

It has been seen that some of the main differences between the PlioMIP1 and PlioMIP2 ensembles are due to the inclusion of new models in PlioMIP2 that were not available at the time of PlioMIP1. We therefore assess whether recent developments in model physics lead to altered responses in Pliocene boundary conditions, in a statistically significant way. In particular, we
assess whether newer models predict a larger Pliocene response than older models. Across the ensemble, model sensitivity to Pliocene boundary conditions does not appear to correlate with the release date of the model (left panels of Supplementary Figure 8; i.e. older models are not demonstrably less sensitive than newer models). An example of this is the GISS2.1G model, which was released in 2019, yet has one of the smallest $Plio_{Core} - PI_{Crtl}$ anomalies for both temperature and precipitation. Within model families, however, some hint of a correlation can be seen. For example, IPSLCM6A (2018) is more sensitive
than IPSLCM5A2 (2017) and IPSLCM5A (2010). The CESM2 model (release date 2020) is more sensitive than the CESM1.2 model (release date 2013), which in turn is more sensitive than CCSM4 (release date 2011), when all are run with the same resolution, boundary and initial conditions. However, CCSM4-Utr (release date 2011) is also very sensitive. This shows that within the CCSM/CESM family, model sensitivity is strongly related to parameterisation choices and initial condition choices, in addition to the release date of the model.

Across the ensemble there is a significant correlation between sensitivity and model resolution (right panels of supplementary Fig. 8), with a larger temperature anomaly and precipitation anomaly predicted in higher resolution models ($p < 0.05$). This suggests that low resolution models may not be able to capture the full extent of climate change shown by higher resolution models. However, it is noted that these relationships are only statistical correlations and some models do not show the same

pattern.  For example, the CCSM4-Utr model has much greater temperature and precipitation anomalies, and CCSM4 has
lower temperature and precipitation anomalies than other models of a similar resolution.

*5.2 Model representations of Pliocene climate vis-a-vis proxy data*

One of the most fundamental changes in experimental design between PlioMIP2/PRISM4 and PlioMIP1/PRISM3D was the
approach towards geological data synthesis for data/model comparison, in particular, moving from SST and vegetation
estimates for a broad time slab to a short SST time series encompassing the MIS KM5c timeslice. This was necessary in order
to assess to what degree climate variability within the Pliocene could affect the outcomes of data/model comparison and,
fundamentally, to derive greater confidence in the outcomes which could be derived from Pliocene data/model comparison
(Haywood et al., 2013a, b). In addition, PlioMIP2 contains many new models not used in PlioMIP1, and the PlioMIP2
boundary conditions have changed compared to PlioMIP1 (particularly the Land-Sea Mask, and the topography).
Nevertheless, what emerges from the comparison of the PlioMIP2 SST ensemble to the F&D19_30 SST data set is a nuanced
picture of widespread model/data agreement with specific areas of concern.

Data model comparisons undertaken for PlioMIP1 indicated that the PlioMIP1 ensemble overestimated the amount of SST
change as a zonal mean in the tropics (Dowsett et al., 2012; 2013; Fedorov et al., 2013).  In PlioMIP2 point-based comparisons,
there is little indication of a systematic mismatch between the data and the models.  Models and proxy data appear to be broadly
consistent in the tropics.  The F&D19_30 data set is comprised of alkenone-based SSTs .  In contrast, the PRISM3D data set
used for DMC in PlioMIP1 was time averaged and composed of estimates from a combination of faunal analysis, Mg/Ca and
alkenone-based SSTs. Tierney et al. (2019) demonstrated that the PlioMIP1 ensemble compared well to alkenone-based SST
estimates in the tropical Pacific for the whole mid-Pliocene Warm Period, not just the PlioMIP2 time slice, when the alkenone-
based temperatures were recalculated using the BAYSPLINE calibration. Therefore, the choice of proxy and inter-proxy
calibration alone can be enough to alter the interpretation of the extent to which the model and data agree.   In addition,
comparing the PlioMIP2 results to an additional dataset of published SSTs for the timeslice (McClymont et al., this issue), we
see that the first order outcome of model-data comparison is the same as that shown by the comparison to F&D19_30.

The Pliocene minus pre-industrial SST anomaly will not only depend on which SST dataset is chosen to represent the Pliocene,
but also on the choice of observed SST data set used for the pre-industrial.  Supplementary Fig. 9 shows the proxy data
reconstructed SST change using the F&D19_30 data set but using two different observed data sets for pre-industrial SSTs to
create the required proxy data SST anomaly.  Using recently released NOAA ERSST V5 data set (Huang et al., 2017) to create
the anomaly instead of the older HadISST data (Rayner et al., 2003) leads to three sites in the North Atlantic showing a much-
reduced Pliocene warming.  It also means that several sites in the tropics now show a small (2 to 3°C) warming during the
Pliocene, while using HadISST data led to an absence of SST warming at these locations. The difference between using NOAA

ERSST V5 or HadISST is sufficiently large that it can determine whether the PlioMIP2 ensemble is able to largely match (or mismatch) the proxy-reconstructed temperatures.

Another region of data/model mismatch noted in PlioMIP1 was the North Atlantic Ocean (NA). Haywood et al. (2013a) noted a difference in the model-predicted (multi-model mean) versus proxy reconstructed (PRISM3D) warming signal of between 2 to 7°C in the NA. The PlioMIP2 multi-model mean SST change appears to be broadly consistent with the F&D19_30 data set,

with a SST anomaly at two sites matching to within 1°C and the other to within 3°C (Fig. 8). There are several possible ways to account for this apparent improvement. Firstly, the total number of sites in the NA in F&D19_30 is reduced compared to the PRISM3D SST data set (Dowsett et al. 2010). The site that led to the 7°C difference noted in Haywood et al. (2013a) is not present in the F&D19_30 data set. Secondly, the PlioMIP2 experimental design specified both the Canadian Archipelago and Bering Strait as closed. Otto-Bliesner et al. (2017) performed a series of sensitivity tests based on the NCAR CCSM4

PlioMIP1 experiment and found the closure of these Arctic gateways strengthened the AMOC by inhibiting transport of less saline waters from the Pacific to the Arctic Ocean and from the Arctic Ocean to the Labrador Sea, leading to warming of NA SSTs. Dowsett et al. (2019) also demonstrated an improved consistency between the proxy-based SST changes and model-predicted SST changes after closing these Arctic gateways in models. It is therefore likely that the multi-model mean SST change in the NA in PlioMIP2 has been influenced by the specified change in Arctic gateways leading to a regionally enhanced

fit with proxy data. However, the question regarding the veracity of the specified changes in Arctic gateways in the PRISM4 reconstruction, given the uncertain and lack of geological evidence either way remains open and requires further study.

One of the clearest data/model inconsistencies occurs in the Benguela upwelling system, where proxy data indicates more SST warming than the multi-model mean. The simulation of upwelling systems is particularly challenging for global numerical climate models due to the spatial scale of the physical processes involved, and the capability of models to represent changes

in the structure of the water column (thermocline depth) as well as cloud/surface temperature feedbacks. Dowsett et al. (2013) noted SST discrepancies between the PRISM3D SST reconstruction and the PlioMIP1 ensemble. Their analysis of the seasonal vertical temperature profiles from PlioMIP1 for the Peru Upwelling region indicated that models produced a simple temperature offset between PI and the Pliocene but did not simulate any change to thermocline depth.

An assumption that proxy-data truly reflect mean annual SSTs in upwelling regions is also worthy of consideration. In

upwelling zones, nutrients (and relatively cold waters) are brought to the surface increasing productivity. The upwelling of nutrient rich waters is often seasonally modulated, which could conceivably bias alkenone-based SSTs to the seasonal maximum for nutrient supply and therefore coccolithophore productivity and/or alkenone flux. In the modern ocean, across the most intense region of Benguela upwelling, the productivity seems to be year-round, whereas the southern Benguela has highest productivity during the summer (Rosell-Melé and Prahl, 2013). Ismail et al. (2015), based on observational data,

demonstrated that it was surface heating, not vertical mixing related to upwelling, which controls the upper ocean temperature gradient in the region today. This lends some credence to the idea that the observed mismatch between PlioMIP2 ΔSST and the F&D19_30 proxy-based anomaly could arise from the complexities/uncertainties associated with interpreting alkenone-

based SSTs in the region as simply an indication of mean annual SST (Leduc et al. 2014). However, we note that no seasonal bias has been identified in the modern dataset in the Benguela region (Tierney and Tingley, 2018).


*5.3 Equilibrium Climate Sensitivity, Earth System Sensitivity and Pliocene climate*

From the analysis shown in section 3.6 a strong relationship between ECS and the ensemble-simulated Pliocene temperature anomaly is discernible. This point is true for the globally averaged temperature anomaly, latitudinal average temperature anomalies in the tropics and specific gridbox based temperature anomalies over large portions of the globe. Across the

ensemble, the tropical Pliocene temperature anomaly is more strongly related to ECS than other latitudes, both as a latitudinal mean and also when considering individual gridpoints. On a gridpoint by gridpoint basis, the tropical oceans are strongly related to modelled ECS, suggesting that SST data from the Pliocene tropics has the potential to constrain model estimates of ECS, highlighting the benefits for deriving estimates of ECS from a concentrated effort to reconstruct tropical SST response using the geological record.

For PlioMIP1, Hargreaves and Annan (2016) also found that modelled $Plio_{Core} - PI_{Crtl}$ SST anomalies over the tropics (30°N-30°S) were correlated with modelled ECS, according to:

$$ECS = \alpha\, \Delta T(30°N - 30°S) + C + \varepsilon \qquad (2)$$

Where α and C are constants, and ε represents all errors in the regression equation. They then used equation (2) along with tropical SST data from PRISM3D (an interpolated dataset of Pliocene proxy SST) to provide a Pliocene data constrained estimate of ECS of 1.9°C – 3.7°C. In order to constrain ECS from the data and modelling used in PlioMIP2, we slightly amend the Hargreaves and Annan (2016) methodology because PlioMIP2 proxy data is more sparsely distributed than PlioMIP1 proxy data and we cannot obtain a reliable estimate of tropical average SST from the data available. To estimate ECS for

PlioMIP2 we instead rely on point-based observations (Fig. 8a) and local regressions between $Plio_{Core} - PI_{Crtl}$ SST and modelled ECS (Figure 7c). Hence, we apply equation (2) with ΔSST from individual data sites, and α and C will now be location dependent. Using this altered methodology, a different estimate of ECS is obtained for each datapoint, these estimates are shown in Fig. 9, and have a range of 2.6°C - 4.8°C with a mean ECS of 3.6°C and a standard deviation of 0.6°C. Fig. 9 does not imply that ECS is different for each location, instead each value in Fig. 9 is an estimate of ECS and incorporates the

true Pliocene constrained ECS along with several errors. For a data-point to be included in Fig. 9 we required that two conditions were met. Firstly, we required that the relationship between local $Plio_{Core} - PI_{Crtl}$ and a model's ECS was significant at the 95% confidence level ($p < 0.05$; these regions are hatched in Fig. 7d). Secondly, we required that at least one of the models in the PlioMIP2 ensemble was within 1°C of the data; this second condition meant that we excluded two sites off the

Eastern United States, two sites from the Mediterranean, and two site from Benguela – despite these sites showing a theoretical relationship between $Plio_{Core} - PI_{Crtl}$ and ECS. Altogether 13 datapoints fulfilled both these conditions and could be used to estimate ECS. The range of estimates of ECS from PlioMIP2 (2.6°C-4.8°C) are similar to IPCC (1.5°C – 4.5°C) but are slightly larger than was estimated from PlioMIP1 (1.9°C -3.7°C). It is not currently possible to add reliable error bars to the range of ECS estimates from PlioMIP2. However, as the Tier1 PlioMIP2 experiments with $CO_2$ set to 350ppmv and 450ppmv become available we will be able to provide an indication as to how uncertainties in the KM5c $CO_2$ would affect the PlioMIP2/PRISM4 constrained estimates of ECS. In addition, as more orbitally tuned SST data becomes available, it will be necessary to revisit the ECS analysis in order to ensure maximum accuracy.

The emergence of the concept of longer-term sensitivity, ESS, can be at least partly attributed to the study of the Pliocene epoch (Lunt et al. 2010; Haywood et al., 2013a). However, as Hunter et al. (2019) state clearly, the comparison between ECS, $Plio_{Core} - PI_{Crtl}$, and ESS can only be robust if an assumption is made that the PlioMIP2 model boundary conditions are a good approximation to the equilibrated Earth system with enhanced atmospheric $CO_2$ concentration. This may appear to be a reasonable assumption now, since the changes in non-glacial elements of the PRISM4 palaeogeography are limited. Yet, within the bounds of plausible uncertainty, a larger number of additional palaeogeographic modifications remain possible for the Late Pliocene than were incorporated into the PRISM4 reconstruction (see Hill 2015 and De Schepper et al., 2015), and which may have a bearing on how well the Pliocene is seen to approximate an equilibrated modern Earth system in the years ahead.

PlioMIP1 determined a range in the ESS/CS ratio of between 1.1 and 2.0, with a best estimate of 1.5. In PlioMIP2, which has benefited from the access to a larger array of models and new boundary conditions, the range in and best estimate for the ESS/CS ratio is similar but slightly larger (1.1 to 2.9 and 1.7 respectively). Therefore, new modelling and new constraints on the data for PlioMIP2 suggests a slight increase in estimates of both the ESS/CS ratio and data constrained estimates of ECS between PlioMIP1 and PlioMIP2.

## 6. Conclusions

The Pliocene Model Intercomparison Project Phase 2 represents one of the largest ensembles of climate models of different complexities and spatial resolution ever assembled to study a specific interval in Earth history. PlioMIP2 builds on the findings of PlioMIP1 and incorporates state-of-the-art reconstructions of Pliocene boundary conditions and new temporally consistent sea-surface temperature proxy data which underpins the new data/model comparison. The major findings of the work include:

- Global annual mean surface air temperatures increases by 1.7 to 5.2°C compared to the pre-industrial, with a multi-model average increase of 3.2°C.
- The multi-model mean annual total precipitation rate increases by 7% compared to the pre-industrial, while the modelled range of precipitation increases by between 2% and 13%.

- The multi-model mean anomaly between Pliocene and pre-industrial is statistically robust for surface air temperature and sea surface temperature over most of the globe. The multi-model mean precipitation anomaly is robust at mid-high latitudes and in monsoon regions but is smaller than inter-model standard deviation in many parts of the tropics and subtropics.

- The degree of polar amplification of surface air temperature change is generally consistent with RCP transient climate modelling experiments used to predict future climate, implying that $CO_2$ changes dominate the ice sheet changes in the $Plio_{Core}$ experiments.

- The land warms more than the oceans in a manner akin to future climate change simulations.

- As an ensemble, average NH warming does not show a clear seasonal cycle, but a clear seasonal cycle is seen in many individual models.

- The difference in the average warming between the hemispheres is subdued, relative to simulations of 2100 CE climate. This is likely due to the substantial changes to the albedo feedback mechanism in the Southern Hemisphere following the removal of large areas of the Antarctic ice sheet in the mid-Pliocene.

- There is a statistically significant relationship between ECS and Pliocene global annual average temperature change. The PlioMIP2 ensemble finds that ESS is greater than ECS by a best estimate of 67%.

- Model estimates of the relationship between ECS and $Plio_{Core} - PI_{Crtl,}$ combined with the PlioMIP2 ΔSST, provides a data constrained estimate of ECS with a range of 2.6°C-4.8°C. This is larger than the values suggested from PlioMIP1 (1.9°C -3.7°C).

- Across the ensemble, there is no clear relationship between the simulated temperature and precipitation anomalies and the year of model release. However newer models may be more sensitive than older models within the same 'family'.

- The PlioMIP2 model ensemble shows broad agreement on polar amplification of the global warming signal and tropical enhancement of rainfall anomalies. Inter-model differences in simulated temperature are mostly found in polar regions and where land-sea-mask and orography of Pliocene paleogeography differ from today.

- The PlioMIP2 ensemble appears to be broadly reconcilable with new temporally specific records of sea surface temperatures. Significant agreement between simulated and reconstructed temperature change is seen, with notable local signals of data/model disagreement. Differences between observed pre-industrial sea surface temperature data sets are large enough to have a significant impact on how well models reproduce proxy-reconstructed ocean temperature changes.

**Acknowledgments**

We acknowledge the use of NOAA_ERSST_V5 data provided by the NOAA/OAR/ESRL PSD, Boulder, Colorado, USA, from their web site at https://www.esrl.noaa.gov/psd/. AMH, JCT, AMD, SJH and DJH, acknowledge the FP7 Ideas: European Research Council (grant no. PLIO-ESS, 278636), the Past Earth Network (EPSRC grant no. EP/M008.363/1) and the University of Leeds Advanced Research Computing service. JCT was also supported through the Centre for Environmental Modelling And Computation (CEMAC), University of Leeds. HJD and KMF acknowledge support from

the USGS Climate Research and Development Program. This research used samples and/or data provided by the Ocean Drilling Program (ODP) and International Ocean Discovery Program (IODP). Any use of trade, firm, or product names is for descriptive purposes only and does not imply endorsement by the U.S. Government. BLO-B and RF acknowledge that material for their participation is based upon work supported by the National Center for Atmospheric Research, which is a major facility sponsored by the National Science Foundation (NSF) under Cooperative Agreement No. 1852977 and NSF OPP grant 1418411. The CESM project is supported primarily by the National Science Foundation. Computing and data storage resources, including the Cheyenne supercomputer (doi:10.5065/D6RX99HX), were provided by the Computational and Information Systems Laboratory (CISL) at NCAR. NCAR is sponsored by the National Science Foundation. NT, CC and GR were granted access to the HPC resources of TGCC under the allocations 2016-A0030107732, 2017-R0040110492 and 2018-R0040110492 (gencmip6) and 2019-A0050102212 (gen2212) provided by GENCI. The IPSL-CM6 team of the IPSL Climate Modelling Centre (https://cmc.ipsl.fr) is acknowledged for having developed, tested, evaluated, tuned the IPSL climate model, as well as performed and published the CMIP6 experiments. CS acknowledges funding by the Helmholtz Climate Initiative REKLIM. CS and GL acknowledge funding via the Alfred Wegener Institute's research programme Marine, Coastal and Polar Systems. QZ acknowledge support from the Swedish Research Council (2013-06476 and 2017-04232). Simulations with EC-Earth were performed on resources provided by the Swedish National Infrastructure for Computing (SNIC) at the National Supercomputer Centre (NSC). WLC and AAO acknowledge funding from JSPS KAKENHI grant 17H06104 and MEXT KAKENHI grant 17H06323. Their simulations with MIROC4m were performed on the Earth Simulator at JAMSTEC, Yokohama. The work by AvdH and MLJB was carried out under under the program of the Netherlands Earth System Science Centre (NESSC), financially supported by the Ministry of Education, Culture and Science (OCW, grant #. 024.002.001). Simulations with CCSM4-Utr were performed at the SURFsara dutch national computing facilities and were sponsored by NWO-EW (Netherlands Organisation for Scientific Research, Exact Sciences) under the project 17189. WRP and DC were supported by Canadian NSERC Discovery Grant A9627 and they wish to acknowledge the support of SciNet HPC Consortium for providing computing facilities. SciNet is funded by the Canada Foundation for Innovation under the auspices of Compute Canada, the Government of Ontario, the Ontario Research Fund – Research Excellence, and the University of Toronto. XL acknowledges financial support from China Scholarship Council (201804910023) and the China Postdoctoral Science Foundation funded project (2015M581154). The NorESM simulations benefitted from resources provided by UNINETT Sigma2 – the National Infrastructure for High Performance Computing and Data Storage in Norway.

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

| (a) Model ID, Vintage | (b) Sponsor(s), Country | (c) Atmosphere Top Resolution and Model References | (d) Ocean* Resolution Vertical Coord., Top BC, & Model References | (e) Sea Ice* Dynamics, Leads & Model References | (f) Coupling* Flux adjustments and Model References | (g) Land Soils, Plants, Routing & Model References | (h) PlioMIP2 Experiment Eoi400 (Boundary Conditions & Experiment Citation) | (i) Vegetation (Static - Salzmann et al. 2008 or Dynamic) | (j) Climate Sensitivity (ECS) °C (incl. source) |
|---|---|---|---|---|---|---|---|---|---|
| CCSM4 (CESM 1.0.5) 2011 | National Center for Atmospheric Research | Top = 2 hPa FV0.9x1.25 (~1°), L26 (CAM4) (Neale et al. 2010a) | G16 (~1°), L60 depth, rigid lid | Rheology, melt ponds Holland et al. (2012); Hunke and Lipscomb (2010) | No adjustments Gent et al. (2011) | Layers, prescribed vegetation type with prognostic phenology, carbon cycle, routing Oleson et al. (2010) | Enhanced Feng et al. 2020 (in review) | Salzmann et al. (2008) | 3.2 (Bitz et al. 2012) |
| CCSM4_ Utrecht (CESM 1.0.5) 2011 | IMAU, Utrecht University, the Netherlands | As CCSM4 except FV (2.5°x 1.9°) | As CCSM4 but with parameterisation changes described in section 3.1 | as CCSM4 | CPL7 Craig et al. (2012) | as CCSM4 | Enhanced | Salzmann et al. (2008) | 3.2 (Baatsen et al, in prep) |
| CCSM4-UoT 2011 | University of Toronto, Canada | As CCSM4 | As CCSM4 but with parameterisation changes described in section 3.1 | as CCSM4 | As CCSM4 | as CCSM4 | Enhanced Chandan and Peltier (2017, 2018) | Salzmann et al. (2008) | 3.2 (Chandan and Peltier, 2018) |
| CESM1.2 2013 | National Center for Atmospheric Research | Top = 2 hPa FV0.9x1.25 (~1°), L30 (CAM5) (Neale et al. 2010b) | G16 (~1°), L60 depth, rigid lid | as CCSM4 | No adjustments Hurrell et al. (2013) | as CCSM4 | Enhanced Feng et al. 2020 (in review) | Salzmann et al. (2008) | 4.1 (Gettelman et al. 2012) |
| CESM2 2020 | National Center for Atmospheric Research | Top = 2 hPa FV0.9x1.25 (~1°), L32 (CAM6) Danabasoglu et al. (2020) | G16 (~1°), L60 depth, rigid lid, updated mixing scheme | Rheology, melt ponds, mushy physics (Hunke et al., 2015) | No adjustment Danabasoglu et al. (2020) | Layers, prescribed vegetation type with prognostic phenology, carbon and nitrogen cycle, routing (Lawrence et al., 2019) | Enhanced Feng et al. (2020, in review) | Salzmann et al. (2008) | 5.3 Gettelman et al. (2019) |

| (a) Model ID, Vintage | (b) Sponsor(s), Country | (c) Atmosphere Top Resolution and Model References | (d) Ocean* Resolution Vertical Coord., Top BC, & Model References | (e) Sea Ice* Dynamics, Leads & Model References | (f) Coupling* Flux adjustments and Model References | (g) Land Soils, Plants, Routing & Model References | (h) PlioMIP2 Experiment Eoi400 (Boundary Conditions & Experiment Citation) | (i) Vegetation (Static - Salzmann et al. 2008 or Dynamic) | (j) Climate Sensitivity (ECS) °C (incl. source) |
|---|---|---|---|---|---|---|---|---|---|
| COSMOS COSMOS-landveg r2413 2009 | Alfred Wegener Institute, Germany | Top = 10 hPa T31 (3.75⍰⍰x 3.75⍰), L19 Roeckner et al. (2003) | Bipolar orthogonal curvilinear GR30, L40 (formal 3.0⍰⍰x 1.8⍰) Depth, free surface Marsland et al. (2003) | Rheology, leads Marsland et al. (2003), | No adjustments Jungclaus et al. (2006) | Layers, canopy, routing Raddatz et al. (2007), Hagemann and Dümenil (1998), Hagemann and Gates (2003) | Enhanced Stepanek et al. (in prep.) | Dynamic | 4.7 Stepanek et al. (2020) |
| EC-Earth 3.3 2019 | Stockholm University, Sweden | IFS cycle 36r4 Top = 5 hPa 1.125° x 1.125°, L62 Döscher et al. (2020) | NEMO3.6, ORAC1 1.0° x 1.0°, L46 Madec (2008) | LIM3 Vancoppenolle et al. (2009) | No adjustments Hazeleger et al. (2012) | Layers, canopy, routing Balsamo et al. (2009), Balsamo et al. (2011) | Enhanced Zheng et al. (2019) | Salzmann et al. (2008) | 4.3 Wyser et al. (2020) |
| GISS2.1G 2019 | Goddard Institute for Space Studies, USA | Top = 0.1 mb 2.0° x 2.5°, L40 Kelley et al. (in prep) | 1.0˚ x 1.25˚, L40 P*, free surface Kelley et al. (in prep) | Visco-plastic rheology, leads, melt ponds Kelley et al. (in prep) | No adjustments Kelley et al. (in prep) | Layers, canopy, routing Kelley et al. (in prep) | Enhanced Chandler et al. (in prep) | Salzmann et al. (2008) | 3.3 (Kelley et al. in prep) |
| HadCM3 1997 | University of Leeds, United Kingdom | Top = 5 hPa 2.5° x 3.75°, L19 Pope et al. (2000) | 1.25° x 1.25°, L20 Depth, rigid lid Gordon et al. (2000) | Free drift, leads Cattle and Crossley, (1995) | No adjustments Gordon et al. (2000) | Layers, canopy, routing Cox et al. (1999) | Enhanced Hunter et al. (2019) | Salzmann et al. (2008) | 3.5 Hunter et al. (2019) |
| IPSLCM6A-LR 2018 | Laboratoire des Sciences du Climat et de l'Environnement (LSCE), France | Top = 1 hPa 2.5° x 1.26°, L79 Hourdin et al. (in prep) | 1° x 1°, refined at 1/3° in the tropics, L75 Free surface, Z-coordinates Madec et al. (2017) | Thermodynamics, Rheology, Leads Vancoppenolle et al. (2009), Rousset et al. (2015) | No adjustments Marti et al. (2010), Mignot et al. (in prep) | Layers, canopy, routing, phenology Peylin et al. (in prep) | Enhanced Contoux et al. (in-prep) | Salzmann et al. (2008) | 4.8 Mignot et al. (in prep) |
| IPSLCM5A2.1 2017 | Laboratoire des Sciences du Climat et de l'Environnement (LSCE), France | Top = 70 km 3.75° x 1.9°, L39 Hourdin et al. (2006, 2013), Sepulchre et al. (in prep) | 0.5°-2° x 2°, L31 Free surface, Z-coordinates Dufresne et al. (2013), Madec et al. (1996), Sepulchre et al. (in prep) | Thermodynamics, Rheology, Leads Fichefet and Morales-Maqueda, (1997, 1999), Sepulchre et al. (in prep) | No adjustment Marti et al. (2010), Sepulchre et al. (in prep) | Layers, canopy, routing, phenology Krinner et al., (2005), Marti et al. (2010), Dufresne et al. (2013) | Enhanced Tan et al. (submitted) | Salzmann et al. (2008) | 3.6 Sepulchre Pierre (pers. Comm.) |

| (a) Model ID, Vintage | (b) Sponsor(s), Country | (c) Atmosphere Top Resolution and Model References | (d) Ocean* Resolution Vertical Coord., Top BC, & Model References | (e) Sea Ice* Dynamics, Leads & Model References | (f) Coupling* Flux adjustments and Model References | (g) Land Soils, Plants, Routing & Model References | (h) PlioMIP2 Experiment Eoi400 (Boundary Conditions & Experiment Citation) | (i) Vegetation (Static - Salzmann et al. 2008 or Dynamic) | (j) Climate Sensitivity (ECS) °C (incl. source) |
|---|---|---|---|---|---|---|---|---|---|
| IPSLCM5A 2010 | Laboratoire des Sciences du Climat et de l'Environnement (LSCE), France | Top = 70 km 3.75° x 1.9°, L39 Hourdin et al. (2006, 2013) | 0.5°-2° x 2°, L31 Free surface, Z-coordinates Dufresne et al. (2013), Madec et al. (1996) | Thermodynamics, Rheology, Leads Fichefet and Morales-Maqueda, (1997, 1999) | No adjustment Marti et al. (2010), Dufresne et al. (2013) | Layers, canopy, routing, phenology Krinner et al. (2005), Marti et al. (2010), Dufresne et al. (2013) | Enhanced Tan et al. (submitted) | Salzmann et al. (2008) | 4.1 Dufresne et al. (2013) |
| MIROC4m 2004 | Center for Climate System Research (Uni. Tokyo, National Inst. for Env. Studies, Frontier Research Center for Global Change, JAMSTEC), Japan | Top = 30 km T42 (~ 2.8° x 2.8°) L20 K-1 Developers (2004) | 0.5° -1.4° x 1.4°, L43 Sigma/depth free surface K-1 Developers (2004) | Rheology, leads K-1 Developers (2004) | No adjustments K-1 Developers (2004) | Layers, canopy , routing K-1 Developers (2004); Oki and Sud (1998) | Enhanced Chan et al. (in prep) | Salzmann et al. (2008) | 3.9 (Uploaded 2 x CO₂ minus PI experiment) |
| MRI-CGCM 2.3 2006 | Meteorological Research Institute and University of Tsukuba, Japan | Top = 0.4 hPa T42 (~2.8° x 2.8°) L30 Yukimoto et al. (2006) | 0.5°-2.0° x 2.5°, L23 Depth, rigid lid Yukimoto et al. (2006) | Free drift, leads Mellor and Kantha (1989) | Heat, fresh water and momentum (12°S-12°N) Yukimoto et al. (2006) | Layers, canopy, routing Sellers et al. (1986); Sato et al. (1989) | Standard Kamae et al. (2016) | Salzmann et al. (2008) | 2.8 (Uploaded 2 x CO₂ minus PI experiment) |
| NorESM-F 2017 | NORCE Norwegian Research Centre, Bjerknes Centre for Climate Research, Bergen, Norway | Top = 3.5 hPa 1.9° × 2.5°, L26 (CAM4) | ~1° x 1°, L53 isopycnal layers | Rheology, melt ponds Holland et al., (2012); Hunke and Lipscomb (2010) | No adjustments Gent et al. (2011) | Layers, canopy, routing Lawrence et al. (2012) | Enhanced (modern soils) Li et al. (in prep) | Salzmann et al. (2008) | 2.3 Guo et al. (2019) |

| (a) Model ID, Vintage | (b) Sponsor(s), Country | (c) Atmosphere Top Resolution and Model References | (d) Ocean* Resolution Vertical Coord., Top BC, & Model References | (e) Sea Ice* Dynamics, Leads & Model References | (f) Coupling* Flux adjustments and Model References | (g) Land Soils, Plants, Routing & Model References | (h) PlioMIP2 Experiment Eoi400 (Boundary Conditions & Experiment Citation) | (i) Vegetation (Static - Salzmann et al. 2008 or Dynamic) | (j) Climate Sensitivity (ECS) °C (incl. source) |
|---|---|---|---|---|---|---|---|---|---|
| NorESM-L (CAM4) 2011 | NORCE Norwegian Research Centre, Bjerknes Centre for Climate Research, Bergen, Norway | Top = 3.5 hPa T31 (~3.75° × 3.75°), L26 (CAM4) | G37 (~3° x 3° ), L30 isopycnal layers | Rheology, melt ponds Holland et al., (2012); Hunke and Lipscomb (2010) | No adjustments Gent et al. (2011) | Layers, canopy, routing Lawrence et al. (2012) | Enhanced (modern soils) Li et al. (in prep) | Salzmann et al. (2008) | 3.1 Haywood et al. (2013a) |

**Table 1**: Details of climate models used with the *Plio$_{Core}$* experiment (a to g), plus details of boundary conditions (h), treatment of vegetation (i) and Equilibrium Climate Sensitivity values (j) (°C).

1125

| Model Name | ECS | Eoi400 SAT | E280 SAT | Eoi400-E280 SAT | ESS (eqn 1) | ESS/CS Ratio |
|---|---|---|---|---|---|---|
| CCSM4-Utrecht | 3.2 | 18.9 | 13.8 | 4.7 | 9.1 | 2.85 |
| CCSM4 | 3.2 | 16.0 | 13.4 | 2.6 | 5.1 | 1.59 |
| CCSM4-UoT | 3.2 | 16.8 | 13.0 | 3.8 | 7.3 | 2.29 |
| CESM1.2 | 4.1 | 17.3 | 13.3 | 4.0 | 7.7 | 1.89 |
| CESM2 | 5.3 | 19.3 | 14.1 | 5.2 | 10.0 | 1.88 |
| COSMOS | 4.7 | 16.9 | 13.5 | 3.4 | 6.5 | 1.39 |
| EC-Earth3.3 | 4.3 | 18.2 | 13.3 | 4.8 | 9.4 | 2.18 |
| GISS2.1G | 3.3 | 15.9 | 13.8 | 2.1 | 4.0 | 1.22 |
| HadCM3 | 3.5 | 16.9 | 14.0 | 2.9 | 5.6 | 1.60 |
| IPSLCM6A | 4.8 | 16.0 | 12.6 | 3.4 | 6.5 | 1.36 |
| IPSLCM5A2 | 3.6 | 15.3 | 13.2 | 2.2 | 4.2 | 1.17 |
| IPSLCM5A | 4.1 | 14.4 | 12.1 | 2.3 | 4.5 | 1.11 |
| MIROC4m | 3.9 | 15.9 | 12.8 | 3.1 | 6.0 | 1.54 |
| MRI-CGCM2.3 | 2.8 | 15.1 | 12.7 | 2.4 | 4.7 | 1.66 |
| NorESM-L | 3.1 | 14.6 | 12.5 | 2.1 | 4.1 | 1.33 |
| NorESM1-F | 2.3 | 16.2 | 14.5 | 1.7 | 3.3 | 1.45 |
| **MMM** | **3.7** | **16.5** | **13.3** | **3.2** | **6.2** | **1.67** |

Table 2: Details of the relationship between the equilibrium climate sensitivity (ECS) and the Earth System Sensitivity (ESS) for each model. MMM denotes the multi-model mean.

| Model name | Root mean squared error (RMSE) | Average difference between data and model | Number of sites where model and data are within 2°C | Number of sites where model and data are within 1°C | Number of sites where model and data are within 0.5°C |
|---|---|---|---|---|---|
| CESM2 | 3.44 | -0.18 | 16 | 9 | 2 |
| IPSLCM6A | 3.38 | 1.17 | 24 | 15 | 8 |
| COSMOS | 3.92 | 1.99 | 20 | 13 | 4 |
| EC-Earth3.3 | 3.34 | -0.45 | 18 | 5 | 1 |
| CESM1.2 | 3.44 | 0.94 | 22 | 13 | 8 |
| IPSLCM5A | 3.83 | 1.76 | 22 | 17 | 6 |
| MIROC4m | 4.05 | 1.95 | 20 | 12 | 9 |
| IPSLCM5A2 | 3.99 | 1.96 | 23 | 17 | 7 |
| HadCM3 | 4.51 | 1.96 | 21 | 13 | 6 |
| GISS2.1G | 4.22 | 2.58 | 19 | 9 | 3 |
| CCSM4 | 4.09 | 2.07 | 21 | 14 | 5 |
| CCSM4-Utr | 3.87 | 0.18 | 19 | 13 | 6 |
| CCSM4-UoT | 3.71 | 1.12 | 21 | 17 | 9 |
| NorESM-L | 4.12 | 2.35 | 21 | 12 | 5 |
| MRI2.3 | 4.78 | 2.13 | 16 | 10 | 8 |
| NorESM1-F | 4.51 | 2.62 | 18 | 10 | 5 |
| **MMM** | **3.72** | **1.51** | **23** | **17** | **10** |

**Table 3: Statistical relationships between the proxy data ΔSST and the model ΔSST at each of the individual grid points. The average difference is calculated as Σ|(SSTA(model) – SSTA(data)| / n, where n is the number of sites.**

# Figures

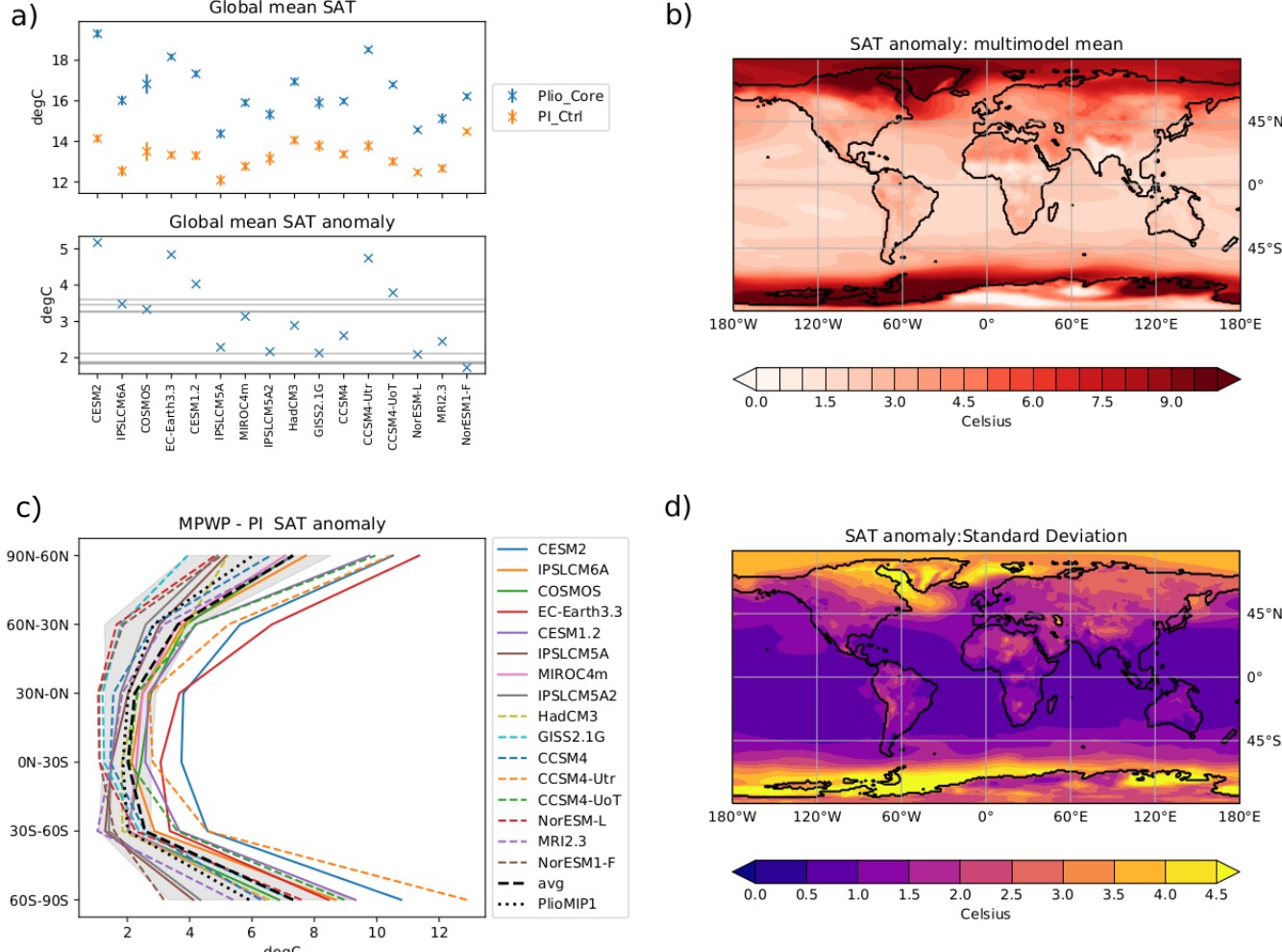

Figure 1: a) Global mean Near Surface Air Temperature [SAT] for the $Plio_{Core}$ and $PI_{Ctrl}$ experiments from each PlioMIP2 model [upper panel] and the difference between them ($Plio_{Core}$ - $PI_{Ctrl}$) [lower panel]. Crosses show the mean value while the vertical bars show the interannual standard deviation. Horizontal grey lines on the lower panel show the anomalies from individual PlioMIP1 models. b) Multimodel mean $Plio_{Core}$ - $PI_{Ctrl}$ SAT anomaly. c) Latitudinal mean $Plio_{Core}$ - $PI_{Ctrl}$ SAT anomaly from each PlioMIP2 model. The PlioMIP2 multimodel mean is shown by the black dashed line. The grey shaded area shows the range of values of the PlioMIP1 models. The PlioMIP1 multimodel mean is shown by the black dotted line. d) Intermodel standard deviation for the $Plio_{Core}$ - $PI_{Ctrl}$ anomaly.

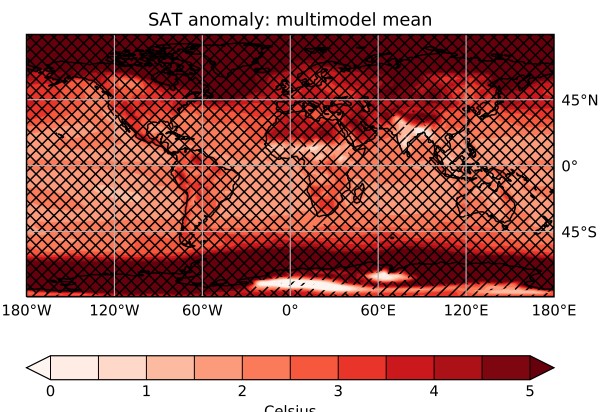

Figure 2: $Plio_{Core}$ - $PI_{Ctrl}$ SAT multimodel mean anomaly. Gridboxes where at least 80% of the models agree on the sign of the change are marked '/'. Gridboxes where the ratio of the multimodel mean SAT change to the $PI_{Ctrl}$ intermodel standard devition is greater than 1 are marked '\'. Gridboxes where both these conditions are satisfied show a robust signal.

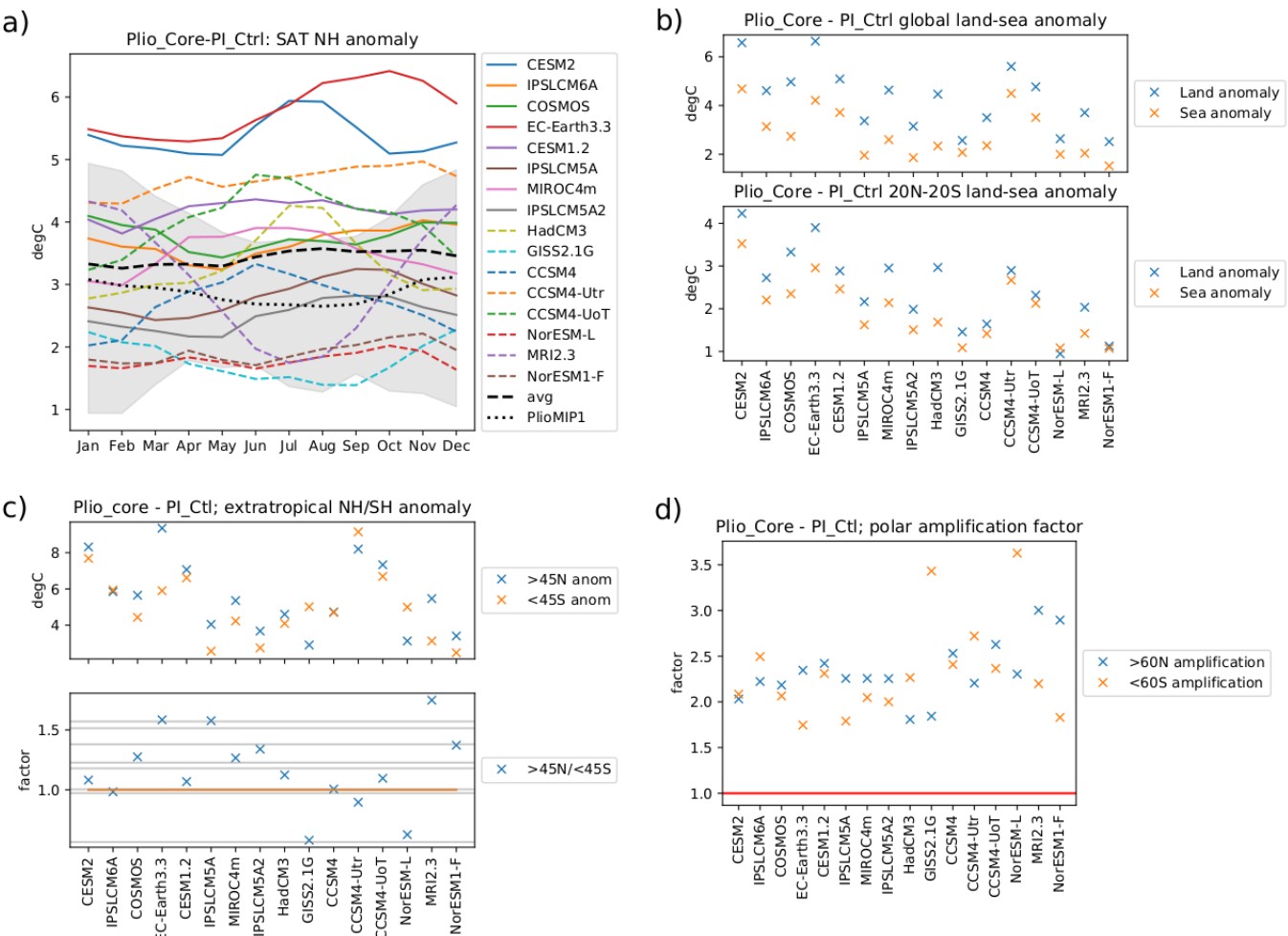

Figure 3: a) Monthly mean NH $Plio_{Core}$ - $PI_{Ctrl}$ SAT anomaly for each PlioMIP2 model, with the PlioMIP2 multimodel mean shown by the black dashed line. The grey shaded region shows the range of values simulated by the PlioMIP1 models and the PlioMIP1 multimodel mean is shown by the black dotted line. b) SAT anomaly for land (blue) and sea (orange) from each model averaged over the globe [top panel] and the 20°N-20°S region [lower panel]. c) SAT anomaly for the northern extratropics (blue) and southern extratropics (orange) [top panel] and the ratio between them [lower panel] The grey horizontal lines on the lower panel show the values from individual PlioMIP1 models. d) SAT anomaly poleward of 60° divided by globally averaged SAT anomaly for the NH (blue) and the SH (orange). The red line highlights a ratio of 1 (i.e. no polar amplification).

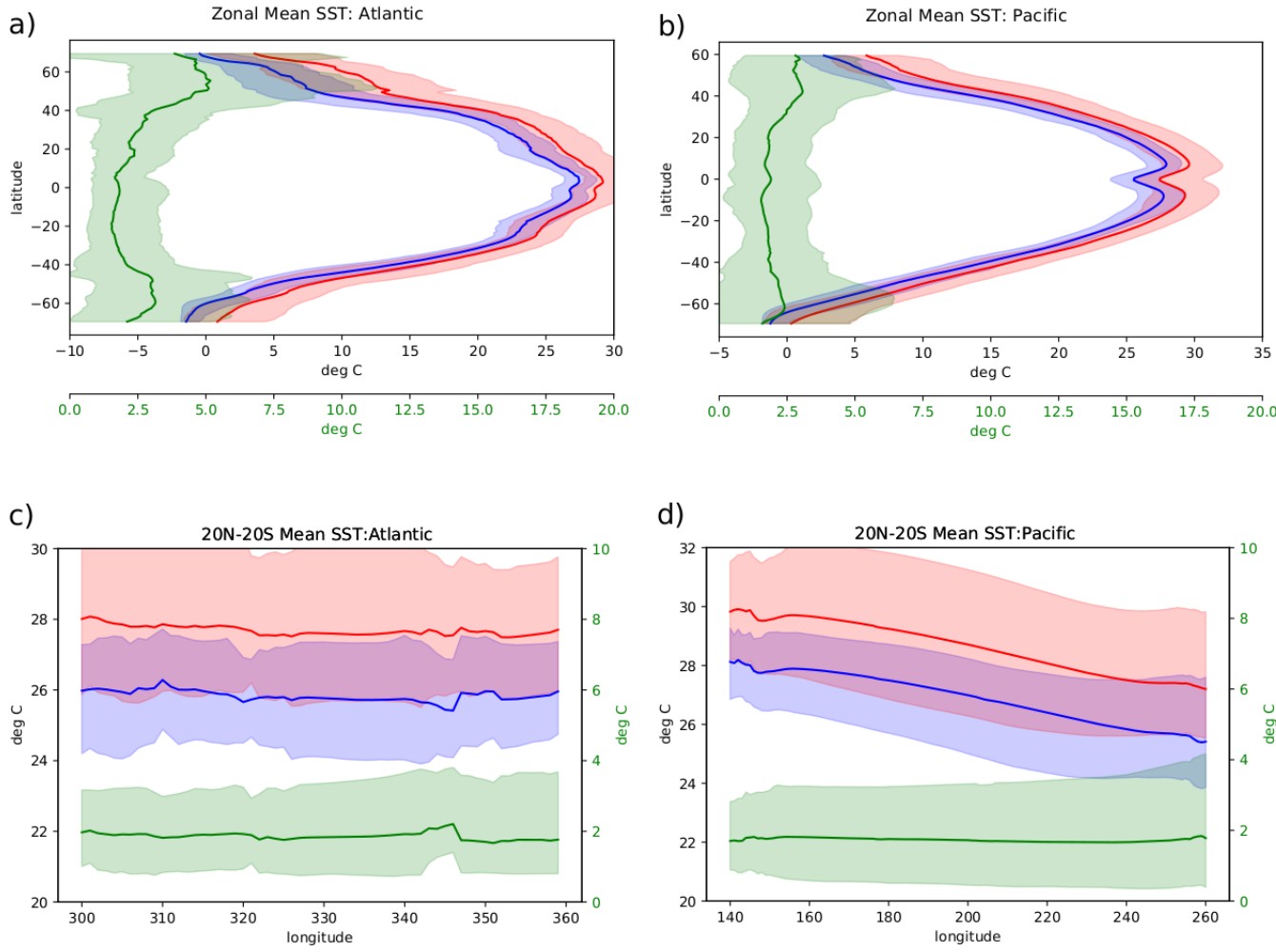

Figure 4: a) and b) show the zonally averaged SST over the Atlantic region (70°W-0°E) and the Pacific region 150°E-100°W respectively. c) and d) show the SST averaged between 20°N and 20°S for the Atlantic and Pacific respectively. In all figures blue shows $PI_{Ctrl}$, red shows $Plio_{core}$ and green shows the anomaly between them. The solid line shows the multimodel mean, while the shaded area shows the range of modelled values.

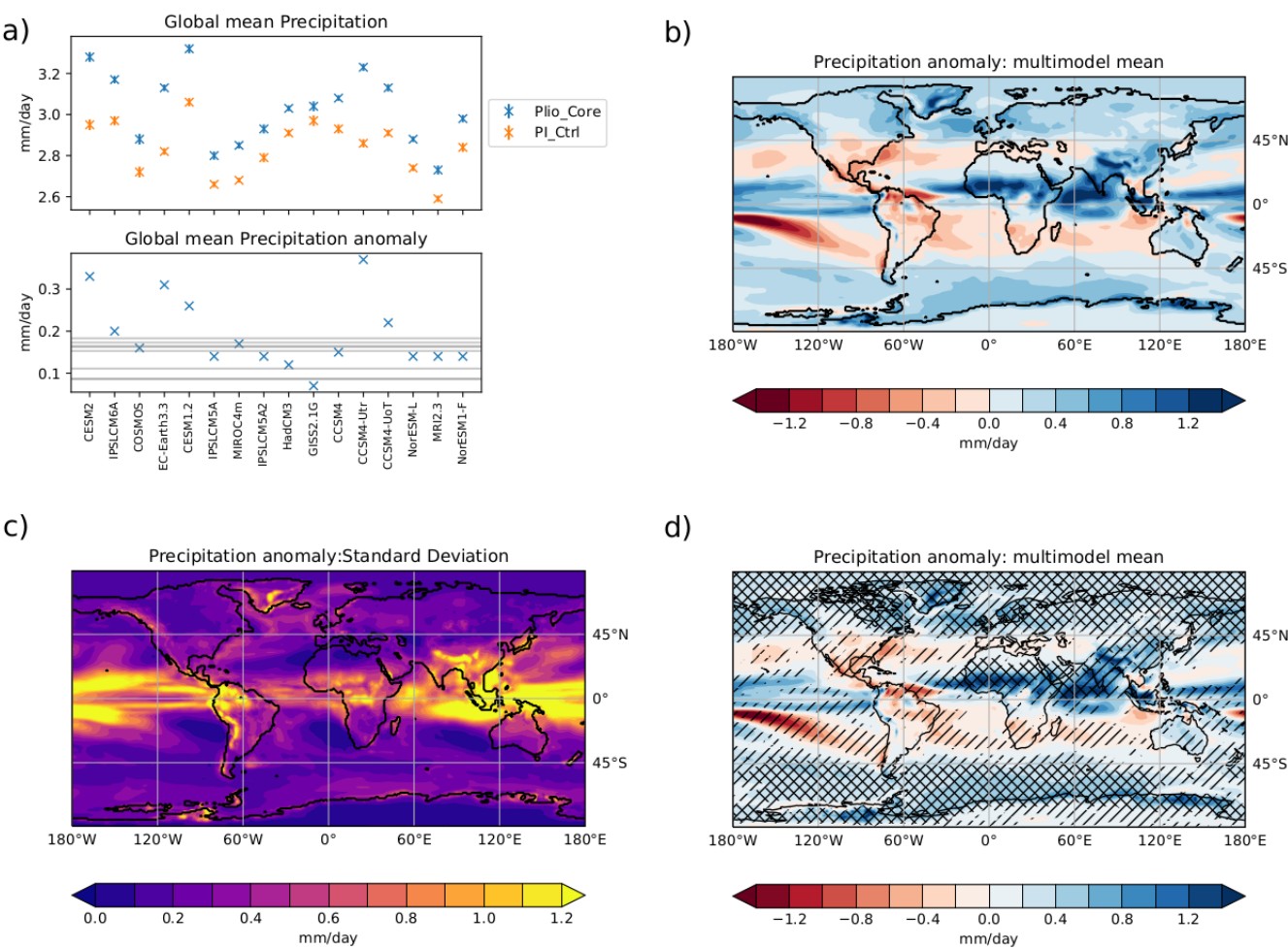

Figure 5: a) Globally averaged precipitation for $Plio_{Core}$ and $PI_{Ctrl}$ from each model [upper panel] and the anomaly between them [lower panel]. The grey horizontal lines on the lower panel show the values that were obtained from each individual PlioMIP1 model b) Multimodel mean $Plio_{Core}$ - $PI_{Ctrl}$ precipitation anomaly. c) Standard deviation across the models for the $Plio_{Core}$ - $PI_{Ctrl}$ Precipitation anomaly. d) $Plio_{Core}$ - $PI_{Ctrl}$ precipitation anomaly, regions which have at least 80% of the models agreeing on the sign of the change are marked with '/'. Regions where the ratio of the multimodel mean precipitation change to the $PI_{Ctrl}$ intermodel standard devition is greater than 1 are marked with '\'.

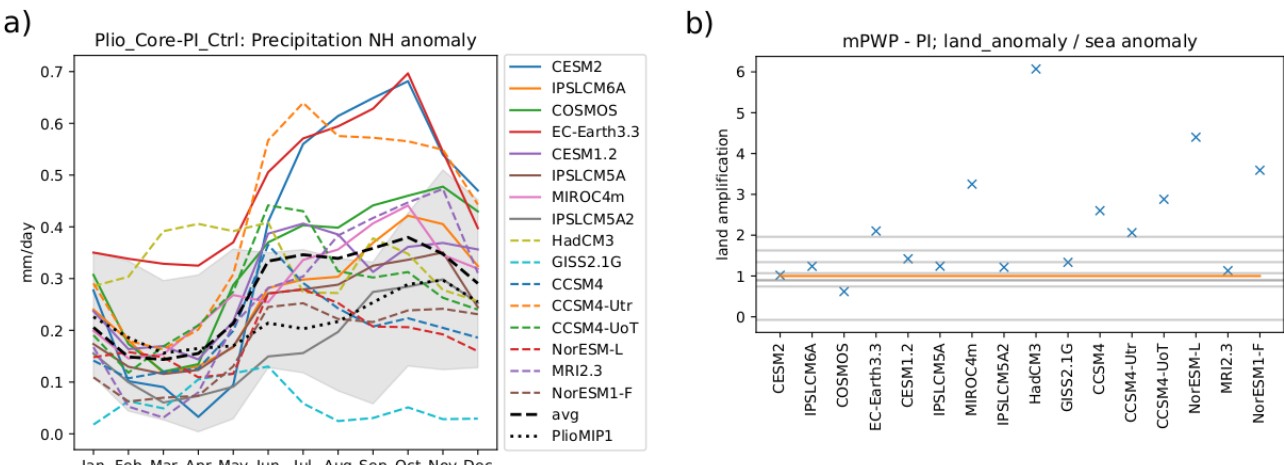

Figure 6: a) shows the NH averaged precipitation anomaly for each PlioMIP2 model and for each month, with the black dashed line showing the PlioMIP2 multimodel mean. The grey shaded region shows the range of values obtained from PlioMIP1 with the PlioMIP1 multimodel mean shown by the black dotted line. b) shows the ratio of the precipitation anomaly over land to the precipitation anomaly over sea for each PlioMIP2 model. Analagous results from individual PlioMIP1 models are shown by the thin grey lines. (A ratio of 1.0 - where the land precipitation anomaly is the same as the sea precipitation anomaly is shown in orange).

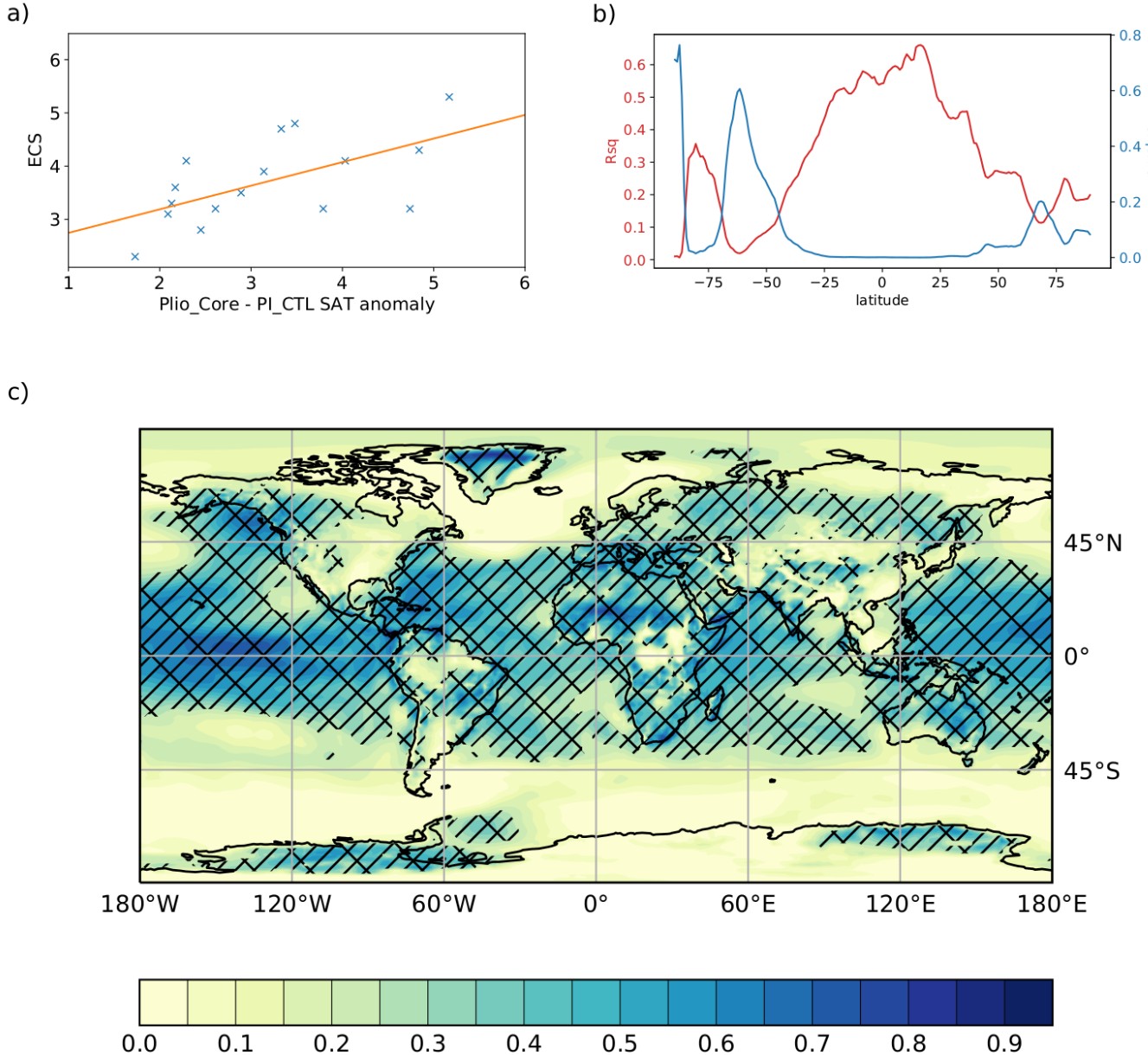

Figure 7: a) the globally averaged $Plio_{Core}$ - $PI_{Ctrl}$ SAT anomaly for each model vs the published equilibrium climate sensitivity (crosses) with the line of best fit shown in orange. b) statistical relationships between the latitudinally averaged $Plio_{Core}$ - $PI_{Ctrl}$ SAT anomaly and the published climate sensitivity. The proportion of climate sensitivity that can be explained by the SAT anomaly at each latitude ($Rsq$) is shown in red, while the probability that there is no correlation between the climate sensitivity and the SAT anomaly ($p$) is shown in blue. c) colors: the percentage variation in modelled ECS that is linearly related to the modelled $Plio_{Core}$ - $PI_{Ctrl}$ SAT anomaly at each gridsquare ($R_{sq}$). Hatching shows a significant relationship (at the 5% confidence level) between the $Plio_{Core}$ - $PI_{Ctrl}$ SAT anomaly at that gridsquare and ECS.

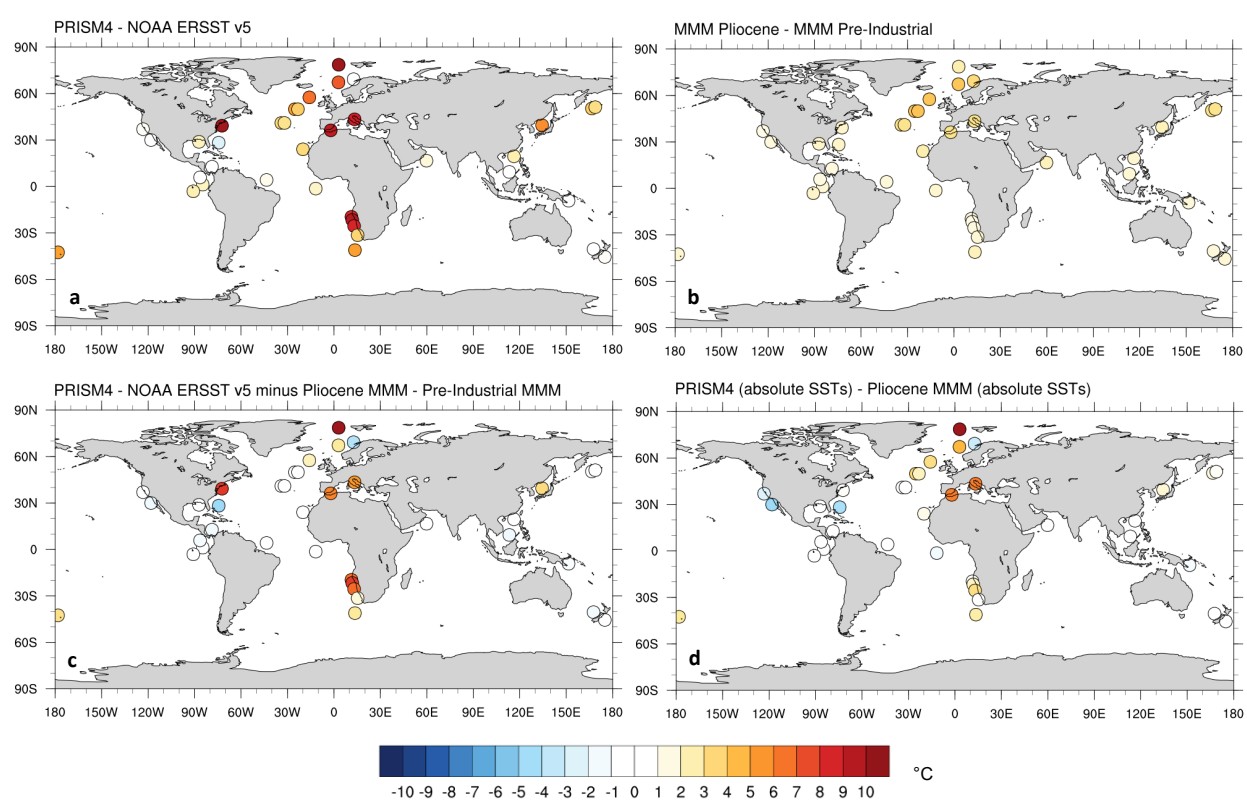

Figure 8: a) PRISM4 - NOAA ERSSTv5 SST anomaly for the datapoints described in section 4. b) multimodel mean $Plio_{Core}$ - $PI_{Ctrl}$ SST anomaly at the points where data are available. c) The difference between the SST anomaly derived from the data (Fig. 8a) and that of the multimodel mean (Fig. 8b). d) the PRISM4 SST data minus the $Plio_{Core}$ multimodel mean.

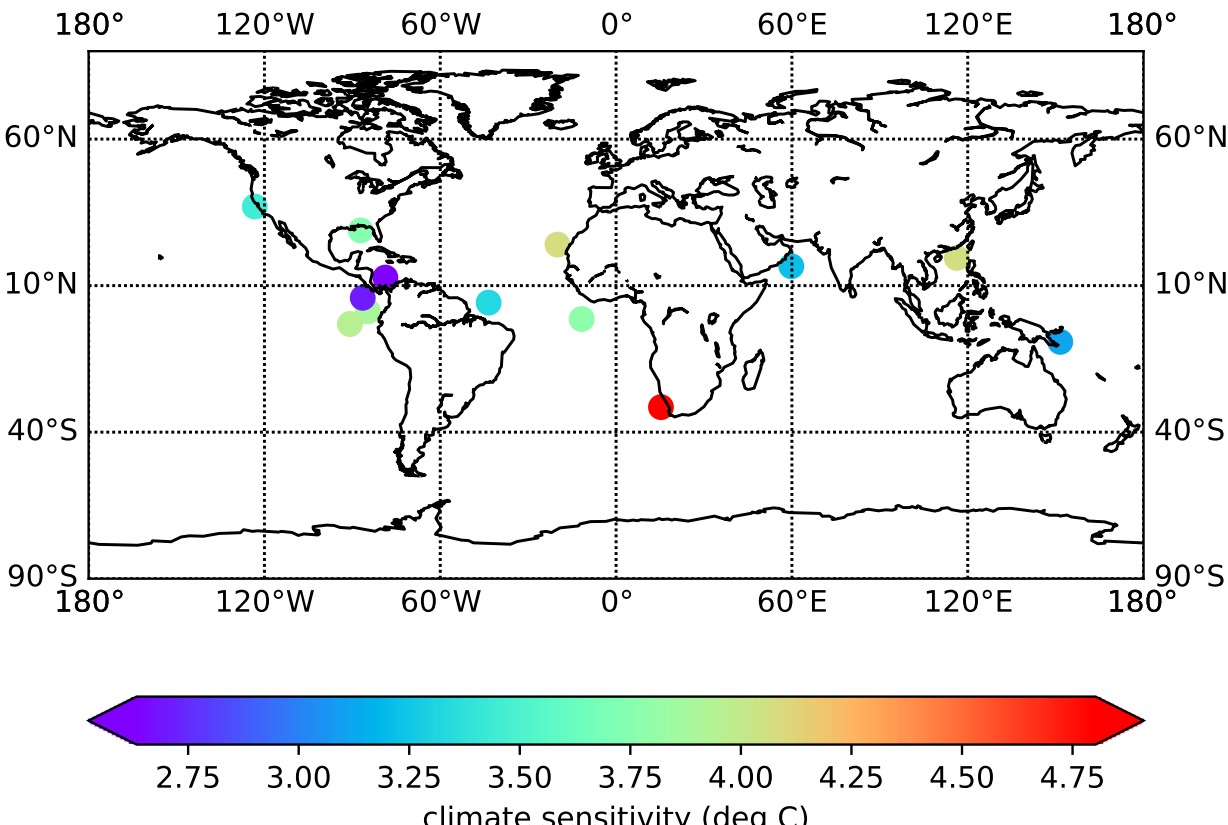

Figure 9: Equlibrium Climate Sensitivity estimated from the data shown in Fig. 8 based on the modelled gridpoint based regressions between $Plio_{Core}$ - $PI_{Ctrl}$ and ECS. This analysis was limited to those data points which were with 1°C of the range of PlioMIP2 models, and was also limited to those sites which were in a gridbox where the modelling suggested a significant relationship between the $Plio_{Core}$ - $PI_{Ctrl}$ SAT anomaly and the ECS.