# Peer review of "The Pliocene Model Intercomparison Project Phase 2: Large scale climate features and climate sensitivity"

_Climate of the Past, 2019_

## Referee Comment (RC2) · Anonymous Referee #2 · 3 Feb 2020

The manuscript by Haywood et al. presents results from the Pliocene Model Intercomparison Project Phase 2 and follows on an earlier Pliocene model intercomparison published in 2013. Despite the title, the manuscript focuses on two issues: Pliocene large-scale climate features and earth system/climate sensitivity. The study has the potential to be an important contribution. As presented, the study has several shortcomings, described below. My recommendation is to revise the manuscript, eliminating the analysis of earth system sensitivity and expanding the analysis of large-scale features and the comparison with proxy data.

General comments:

[Figure]

As an analysis of a model intercomparison project, the manuscript is fine. It reports the ensemble average and range across a spectrum of large-scale features including global, zonal and seasonal temperature; polar amplification; SST gradients, and precipitation rate, and compares them to pre-industrial conditions. These results are interesting and relevant but the analysis is rather cursory. It would have also been interesting for the authors to include some direct comparisons with PlioMIP1 in the Results (for example, include PlioMIP1 ensemble means in the figures and description of results). In the Discussion, there is some speculation about why the PlioMIP2 results differ from PlioMIP1, but it's just speculation. In addition, the analysis of PlioMIP2 models lacks investigation of why large-scale climate features differ among PlioMIP2 models. Both of these are major missed opportunities.

The estimate of earth system sensitivity (ESS) (equation 1) is not explained or justified. There is no a priori reason to think that the ESS will scale as the ratio of ln (560/280)/ln(400/280). This scaling would be appropriate if $CO_2$ were the only factor changing between simulations. That this scaling is inappropriate is illustrated by the differences in ESS for CCSM4 (Table 2, compare CCSM4-2deg and CCSM4-UofT). In these two simulations with the same model, the Eo400-E280 SAT differs by 0.9 C due to difference in treatment of Pliocene boundary conditions. (No surprise.) The ESS estimate (equation 1) grows the difference to 1.8 C. Why would the same model respond so differently to an increase in $CO_2$ of 160 ppm? If the authors believe this result is justified, they must demonstrate it by running these two CCSM4 Pliocene simulations with 560 ppm $CO_2$. Given the shortcoming in the ESS estimate, all discussion of earth system sensitivity should be removed from the manuscript.

Specific comments:

Introduction. The Introduction could be improved. It does a poor job of justifying the rationale for conducting PlioMIP2. A strong case could be made that the PlioMIP2 offers an opportunity to evaluate climate models that have been strongly tuned for the present day, and to showcase advances in modeling since PlioMIP1. Instead,

the Introduction (paragraph 2 specifically) is a clumsily written laundry list of all the publications that resulted from PlioMIP1. What's the point?

L. 156. The minimum integration length was specified as 500 simulated years. For many models, this is not sufficient time to reach equilibrium. I appreciate that the authors report the spin up time and net TOA radiation in Supplementary Figures and suggest that they add this important information to the manuscript.

L. 164. Here and elsewhere (e.g. L. 473) the manuscript mentions the release date of the model, and even makes statements like "the model sensitivity is more strongly related to parameterization choices and initial conditions than the release date of the model". As the authors are well aware, model performance is related to the accurate representation of the dynamics and physics and has nothing to do with the date of release. Please remove these confusing and unnecessary comments.

L. 237. "Lack of consistency in the seasonal signal of warming. . ." An analysis of the global average seasonality is not very useful (or at least ambiguous) here since the seasons are out-of-phase between hemispheres. Please show hemispheric averages, or just the NH average. Also, this is a place where additional analysis would be appreciated to understand the reason for the model differences.

L. 339. ". . .suggests that there are some inconsistencies between the way in which ECS and ESS were obtained." See my comments above about the estimate of ESS used in the manuscript.

L. 344. "each of these models provides a different but equally valid realization of ESS. . ." I don't understand this statement. Please elaborate or delete.

Data/Model Comparison. This section (lines 389-400) focuses on comparing the ensemble mean to the proxy data. How consistent are the models? It would be valuable to add a figure showing model agreement, the number of models that were within the criteria for a good fit. In addition, it would be valuable to estimate and report the goodness of fit for each model, as well as the ensemble mean.

L. 410. Please calculate and report the global mean DSAT/ DSAT estimate from proxy data for comparison. There are a number of ways to do this that have been reported in the literature.

Section 5.1. This section is quite interesting. In line 460-464, it is stated that there are differences between the Pliocene and RCP predictions. Please elaborate on these differences, in the same way that the similarities have been described.

L. 489. "Previous DMCs for the Pliocene. . ." I don't understand this sentence. Please clarify.

L. 545. ". . .suggest that SST data from the Pliocene tropics has the potential to constrain model estimates of ECS. . ." The discussion that follows (L. 550-577) about equilibrium climate sensitivity is not robust. ECS is calculated from a handful of local points from the same regions and is justified because it agrees with the ECS reported in IPCC, the exact value that this analysis should be testing.

L. 629. This is not a conclusion of this study, and therefore shouldn't be included in the Conclusions.

---

## Author Comment (AC1) · 29 Feb 2020

*We would like to thank the reviewer for providing helpful comments on this manuscript. The suggestions will help improve the manuscript in a revised version. We provide a response to specific comments below.*

The manuscript by Haywood et al. presents results from the Pliocene Model Intercomparison Project Phase 2 and follows on an earlier Pliocene model intercomparison published in 2013. Despite the title, the manuscript focuses on two issues: Pliocene large-scale climate features and earth system/climate sensitivity. The study has the potential to be an important contribution. As presented, the study has several shortcomings, described below. My recommendation is to revise the manuscript, eliminating the analysis of earth system sensitivity and expanding the analysis of large-scale features and the comparison with proxy data.

*The manuscript is intended as an introduction to the main results from PlioMIP2 and includes a large range of features in temperature and precipitation across different timescales and spatial scales. We also present a comparison with proxy data and consider the relationship between climate sensitivity and the Pliocene climate anomaly.*

*We agree that it would be possible to move the analysis of Earth System Sensitivity and Climate sensitivity to another paper. However, we do not agree that this is necessary or indeed desirable. The inclusion of this analysis provides a useful link between PlioMIP2 output, Pliocene proxy data and future climate change, which broadens the relevance of the manuscript. Removing this analysis would be to the detriment of the manuscript.*

*However, we acknowledge that the ESS/ECS section can be improved. We will simplify this section in the revised version of the paper. In the previous version complications arose from*

1. *The EC-Earth3.1 model, which had ESS < ECS – which was unreasonable. This model has been now withdrawn from PlioMIP2 by the EC-Earth modelling group due to recently determined problems in sea ice sensitivity.*
2. *The CCSM4 models not being internally consistent. We had three very different representations of the Pliocene climate from CCSM4 due to different groups using different model parameterisations and different ways in which the Pliocene boundary conditions were implemented. We therefore took decision to average these models together to investigate the relationship between ECS (which was the same in all the CCSM4 models) and ESS (which was different in the CCSM4 models).*

*In the revised version of the paper we will include all models in our ECS/ESS analysis, including treating all versions of CCSM4 as separate and distinct models. This will make the ECS/ESS analysis consistent with the rest of the paper, where all versions of CCSM4 were treated as separate models.*

*In addition, since the previous version of the paper we have received two further contributions to PlioMIP2. These are EC-Earth3.3 and CESM2. In the revised version we will include these two additional models, and this strengthen the results seen in the initial submission. In particular, the inclusion of these additional models strengthens the link between ECS and ESS and makes this section more robust.*

*We agree with the reviewer that the title did not accurately reflect the contents of the paper. In the revised version we will change the title to show that the paper deals with both large-scale features of the Pliocene climate and climate sensitivity.*

General comments

As an analysis of a model intercomparison project, the manuscript is fine. It reports the ensemble average and range across a spectrum of large-scale features including global, zonal and seasonal temperature; polar amplification; SST gradients, and precipitation rate, and compares them to pre-industrial conditions. These results are interesting and relevant, but the analysis is rather cursory. It would have also been interesting for the authors to include some direct comparisons with PlioMIP1 in the Results (for example, include PlioMIP1 ensemble means in the figures and description of results). In the Discussion, there is some speculation about why the PlioMIP2 results differ from PlioMIP1, but it's just speculation. In addition, the analysis of PlioMIP2 models lacks investigation of why large-scale climate features differ among PlioMIP2 models. Both of these are major missed opportunities.

*The manuscript represents the first results from the PlioMIP2 ensemble and therefore we included many comparisons/analysis that subsequent studies will expand on. We acknowledge that the reviewer thought the analysis was too brief, however, the manuscript is an overview for a broad audience and already runs to a number of pages, hence we would not like to add a large amount of additional detail.*

*In a number of places, the text compares results with PlioMIP1, however we note from the reviewer's comment that a more quantitative comparison would be desirable. In the revised version of the manuscript we will add the PlioMIP1 results to appropriate figures (fig 1a, fig 1c, fig3, fig4a, fig6).*

*We agree with the reviewer that it would be very interesting to understand why large-scale climate features differ among different models. However, this is a very difficult question to answer and it is beyond the scope of this study to do this thoroughly for all diagnostics. In general, the models that have the greatest Pliocene warming also have the greatest published climate sensitivity, and this will be emphasised more in the conclusions. Where all models have been presented individually (i.e. fig 1a, fig4a etc) we will reorder the models in terms of highest ECS to lowest so that the reader can more clearly see how the diagnostics relate to each other.*

The estimate of earth system sensitivity (ESS) (equation 1) is not explained or justified. There is no a priori reason to think that the ESS will scale as the ratio of ln (560/280)/ln(400/280). This scaling would be appropriate if $CO_2$ were the only factor changing between simulations. That this scaling is inappropriate is illustrated by the differences in ESS for CCSM4 (Table 2, compare CCSM4-2deg and CCSM4-UofT). In these two simulations with the same model, the Eo400-E280 SAT differs by 0.9 C due to difference in treatment of Pliocene boundary conditions. (No surprise.) The ESS estimate (equation 1) grows the difference to 1.8 C. Why would the same model respond so differently to an increase in $CO_2$ of 160 ppm? If the authors believe this result is justified, they must demonstrate it by running these two CCSM4 Pliocene simulations with 560 ppm $CO_2$. Given the shortcoming in the ESS estimate, all discussion of earth system sensitivity should be removed from the manuscript.

*The reviewer makes two points here:*

1. *That equation 1 is not explained or justified*
2. *That the CCSM4-2deg and CCSM4-UoT provide a very different response to the mPWP boundary conditions, and hence ESS (calculated from these different version of the same model) is very different when it should not be.*

*To respond to point 1:*

*the ESS is simply*

*DeltaR \* ESS = DeltaT*
*Where DeltaR is 5.35 \* ln(400/280)*

*As far as the determination of ESS is concerned we are not doing anything unusual. As elaborated in Chandan and Peltier 2018 (this issue), the usage of this equation is justifiable on ESS timescale if we consider the various ice-sheet and GIA related changes as long-timescale earth system responses arising from CO2 changes in the first place. This is illustrated schematically as Figure 11d in that paper. To the extent this argument is correct, i.e. to the extent that non GIA related orography changes are small, and which appears to be the case from Dowsett et al. (2016), the above formula should indeed give an estimate of ESS even though on the LHS the forcing is just that from CO2 and on the RHS the total temperature difference includes contributions from various other factors. A minor caveat is that the non-GIA related changes are most pronounced in the infilling of the straits in Northern Canada and the Hudson Bay. But this would only significantly affect local calculations and is not expected to affect the results from the above equation which is based on a global mean.*

*To respond to point 2:*

*The reviewer notes that "In these two simulations with the same model, the Eo400-E280 SAT differs by 0.9 °C due to difference in treatment of Pliocene boundary conditions. (No surprise.)" Actually, this difference of 0.9°C between two very similar models was a surprise to us, and we spent a great deal of time trying to understand the exact differences between how the boundary conditions had been implemented in CCSM4-UoT and CCSM4-2deg. This is discussed in lines 205-215 of the manuscript. The reviewer's suggestion to run these two CCSM4 simulations with 560ppm CO2 would provide clarity to this issue, however these simulations are very expensive to run and were not a requirement for a modelling group to contribute to PlioMIP2 (Haywood et al. 2016 – this issue).*

*While we agree that some uncertainties remain about the CCSM4 models, we do not agree that this translates into uncertainties of our ESS methodology. The disagreement between CCSM4 models is apparent throughout the manuscript – not just in the ESS section. Even with this disagreement the strong correlation between ESS and ECS is sufficient that the PlioMIP2 ensemble and proxy data can be used to help constrain ECS. This will become more apparent in the revised version due to improvements in the ESS/ECS analysis.*

Specific comments:

 Introduction.

The Introduction could be improved. It does a poor job of justifying the rationale for conducting PlioMIP2. A strong case could be made that the PlioMIP2 offers an opportunity to evaluate climate models that have been strongly tuned for the present day, and to showcase advances in modeling since PlioMIP1. Instead, the Introduction (paragraph 2 specifically) is a clumsily written laundry list of all the publications that resulted from PlioMIP1. What's the point?

*The reviewer makes some good points here. In the revised version of the manuscript we will rewrite the introduction to incorporate the reviewer's suggestions.*

L. 156. The minimum integration length was specified as 500 simulated years. For many models, this is not sufficient time to reach equilibrium. I appreciate that the authors report the spin up time and net TOA radiation in Supplementary Figures and suggest that they add this important information to the manuscript.

*The manuscript has been written to appeal to a broad audience, only some of whom will be interested in the spin up time and the net TOA radiation. We think it is better to keep this information in the supplement, where it can be found by interested parties without distracting other readers.*

L. 164. Here and elsewhere (e.g. L. 473) the manuscript mentions the release date of the model, and even makes statements like "the model sensitivity is more strongly related to parameterization choices and initial conditions than the release date of the model". As the authors are well aware, model performance is related to the accurate representation of the dynamics and physics and has nothing to do with the date of release. Please remove these confusing and unnecessary comments.

*The purpose of this analysis was to determine if developments in model physics lead to altered responses to Pliocene boundary conditions. In particular, whether newer (more recently released) models might show a different sensitivity than older models. We will clarify our meaning in the text.*

L. 237. "Lack of consistency in the seasonal signal of warming..." An analysis of the global average seasonality is not very useful (or at least ambiguous) here since the seasons are out-of-phase between hemispheres. Please show hemispheric averages, or just the NH average. Also, this is a place where additional analysis would be appreciated to understand the reason for the model differences.

*This is a good point. In the revised version of the paper we will just show NH average. We will also remove the seasonal correlation for ESS in figure 7.*

L. 339. "...suggests that there are some inconsistencies between the way in which ECS and ESS were obtained." See my comments above about the estimate of ESS used in the manuscript.

*The EC-Earth3.1 model (about which this statement was written) has been withdrawn from the PlioMIP2 intercomparison due to problems in sea ice sensitivity. This sentence will not be required in an updated version of the manuscript.*

L. 344. "each of these models provides a different but equally valid realization of ESS..." I don't understand this statement. Please elaborate or delete.

*This paragraph will be removed in the revised version of the manuscript. This is because we will be treating all the CCSM4 models as separate models in the ESS section (in the same way we have done in the rest of the manuscript).*

Data/Model Comparison. This section (lines 389-400) focuses on comparing the ensemble mean to the proxy data. How consistent are the models? It would be valuable to add a figure showing model agreement, the number of models that were within the criteria for a good fit. In addition, it would be valuable to estimate and report the goodness of fit for each model, as well as the ensemble mean.

*We agree that additional information about how individual models compare to the proxy data would be useful. In the revised version, a figure showing how each model compares to the proxy data will be added to the supplementary information.*

L. 410. Please calculate and report the global mean DSAT/ DSAT estimate from proxy data for comparison. There are a number of ways to do this that have been reported in the literature.

*This is an interesting suggestion, but the spatial distribution of Pliocene SST data is too limited, in our opinion, to make such a process reliable.*

Section 5.1. This section is quite interesting. In line 460-464, it is stated that there are differences between the Pliocene and RCP predictions. Please elaborate on these differences, in the same way that the similarities have been described.

*Here we were discussing changes in boundary conditions between the PlioMIP2 simulations and RCP simulations. We will make this clear.*

L. 489. "Previous DMCs for the Pliocene..." I don't understand this sentence. Please clarify.

*This sentence will be rewritten in a revised version.*

L. 545. "...suggest that SST data from the Pliocene tropics has the potential to constrain model estimates of ECS..." The discussion that follows (L. 550-577) about equilibrium climate sensitivity is not robust. ECS is calculated from a handful of local points from the same regions and is justified because it agrees with the ECS reported in IPCC, the exact value that this analysis should be testing.

*This discussion arose from two points:*

1. *The Pliocene temperature anomaly at many model gridpoints is correlated with the model's ECS.*
2. *Some of the gridpoints where this correlation occurs have proxy data – providing an independent estimate of the Pliocene climate.*

*While models provide a range of estimates of climate sensitivity, combining points 1 and 2 allow us to estimate climate sensitivity from proxy data at certain locations.*

*We did not intend to justify our method based on it agreeing with IPCC, rather we intended to compare our results with IPCC estimates (and also with earlier PlioMIP1 studies) for completeness. We will rewrite the text around line 571-572 to make this clear.*

*The additional models that have been added to PlioMIP2 since the initial submission improve the relationship between Pliocene temperature anomaly and ECS. This means that the number of proxy data points that can be used to estimate ECS will increase in a revised version, and will cover a wider region.*

L. 629. This is not a conclusion of this study, and therefore shouldn't be included in the Conclusions.

*We agree with this point and this conclusion will be removed in a revised version*

---

## Referee Comment (RC3) · Tim Herbert (Referee) · 30 Mar 2020

zip file attached with:

main review as a .pdf

commented manuscript as a second .pdf

Please also note the supplement to this comment:
https://www.clim-past-discuss.net/cp-2019-145/cp-2019-145-RC3-supplement.zip

---

## Author Comment (AC2) · 16 Apr 2020

We would like to thank Tim Herbert for providing helpful comments on this manuscript, which will help improve a revised version. We provide a response below to those comments that have been incorporated into version 1 of the manuscript:

[TDH1]: This is absolute scaling, but it would be more useful to provide a context: either add the absolute mean SAT, SST, or the scaling between delta SST and delta SAT(land). [line 41]

In the revised version we will specify by how much SAT increases over land and how

much it increases over the ocean.

[TDH2]: Clarify? Meaning of "constraints" not evident. [line 44]

In the previous version there was a statistically significant relationship between a model's Pliocene temperature response and the Equilibrium Climate Sensitivity when we excluded the EC-Earth3.1 model and averaged together the CCSM4 models. These were the modelling constraints we referred to. However, since the previous version of the manuscript the EC-Earth3.1 model has been withdrawn from PlioMIP2 and two further models have been added. This means that we now obtain a statistically significant relationship between a model's Pliocene temperature response and the Equilibrium Climate Sensitivity without any 'modelling constraints'. This sentence is therefore no longer needed in the revised version.

[TDH3]: Data shows polar amplification well equatorward of +-60 so this suggests model not able to capture all the physics? [line 43]

We reported the polar amplification as the ratio of warming poleward of 60° to the global mean warming. This is a standard metric for calculating polar amplification (Smith et al 2019). We do not state that there is no polar amplification equatorward of 60°, and figure 1b also shows that polar amplification occurs equatorward of 60°. Therefore, our paper does not imply model-data disagreement or any problems with model physics in this respect.

[TDH4]: This is all in the context of a constant 400 ppm forcing? E.g. no uncertainty in pCO2? Relevant to deducing ECS from SST- requires pCO2 estimate, correct? [line 48]

This is a good point and in a revised version we will add some further discussion about CO2 uncertainty to the text. The ECS estimate uses two inputs: 1. The data, which we assume comes from a 400ppm CO2 world. However, we note that uncertainties may mean the data represents a world where CO2 was slightly different. 2. The models

which were all run with CO2 of 400ppmv. Given that we do not currently have a range of model simulations with different CO2 values, the only possible way of estimating ECS requires using regressions derived from a CO2 = 400ppmv modelling world. We therefore currently have no other option other than to assume that the data also represents CO2 of 400ppmv. If the data represented a world with a different CO2 value it could not be related to the model outputs. However, the PlioMIP2 model design did accept that there are uncertainties on the KM5c CO2 value (Haywood et al; this issue) and suggested that modelling groups also carried out experiments with CO2 set to 350ppmv and 450ppmv in order to quantify CO2 uncertainties. As the simulations with different CO2 values become available, we will be able to add reliable error bars to our ECS estimates that will account for CO2 uncertainty.

[TDH5]: Of course, given present proxy CO2 data, we don't really know if this window was say +- 20 ppm higher than the canonical 400 ppm. . . [line 115]

Please see response to TDH4.

[TDH6]: I would have thought that reduction in winter sea ice and/or lower land surface albedo would have generated a larger winter warming relative to the mean anomaly. [lines 230-240].

The amount of winter sea ice has very little effect on temperature. This is because there is no sunlight over the winter pole, and so the value of the surface albedo is irrelevant. The reduction in winter snow cover away from the pole will affect only be a small proportion of the northern hemisphere surface. Therefore, it can only have a limited effect on hemisphere averaged temperatures.

[TDH7]: I am a bit mystified here on the Foley/Dowsett data source precision. For example, see Figure 9 of Caballero-Gill et al.: at 3.205 Ma, SST at Site 1125 is ∼21oC and 594 is 14oC. The Foley/Dowsett table gives 19.5 and 12.2 respectively. [lines 382-387]

Figure 9 of Caballero-Gill et al, includes data from 2.6Ma – 4.2Ma and as such data near KM5c only represents a very small proportion of the figure. The figure is therefore not of sufficient temporal resolution to obtain SST directly. The dataset upon which the figure is based for site 1125 (https://doi.pangaea.de/10.1594/PANGAEA.898162) shows that SST does nearly reach 21°C at 3.213Ma, however at 3.205 the SST is 19.5 as reported by Foley and Dowsett. A similar argument could be applied to site 594.

[TDH8]: Suggest moving to McClymont et al. data set in lieu of comments above? [line 397]

Sites which are included in both Foley and Dowsett and McClymont et al. datasets generally show the same or very similar SST estimates. There are some very small differences (0.1-0.2°C) which are undoubtedly due to the different time windows used (10ky and 30ky in Foley and Dowsett, 20ky in McClymont et al.). In some cases, Mc-Clymont et al. used previously unpublished data that could not have been included in Foley and Dowsett. As modellers, we need to validate our models against a wide range of different datasets. Here we choose to validate against the Foley and Dowsett dataset because the model results are already compared with the McClymont dataset elsewhere (McClymont et al., this issue). However, we note that the first order outcomes of the PlioMIP2 model-data comparison is the same, regardless of whether model results are compared with the Foley and Dowsett dataset of the McClymont et al. dataset. This will be noted in a revised version of our manuscript.

[TDH9]: To me, the pattern of data anomalies exceeding model anomalies near the gyre boundaries is quite robust (see MyClymont or example) and is a general feature of "warm climate" reconstructions- see Brierley for an earlier Pliocene time slice, and many others for the Miocene and Eocene. I think there is a fundamental model deficiency here. [line 401-406]

We agree that there could be a fundamental model deficiency in these regions and have stated near line 520 "The simulation of upwelling systems is particularly challenging for global numerical climate models due to the spatial scale of the physical processes involved, and the capability of models to represent changes in the structure of the water column (thermocline depth) as well as cloud/surface temperature feedbacks". However, the interpretation of data in upwelling regions is also not trivial and we also discuss this in our paper.

[TDH10] and [TDH11]: See comments above – I think this SST data set is particularly problematic. [Lines 480-496]

Please see response to TDH7 and TDH8.

[TDH12]: F&D data set is more likely to be the issue. . .[lines 498-503].

Here the reviewer is referring to our analysis which shows that the mPWP-PI temperature anomaly can depend on which dataset we use to represent the preindustrial data. We note that if two different datasets give different values for the PI climate, than these datasets will lead to different values of the mPWP-PI temperature anomaly. This difference will be independent of which dataset was chosen to represent the mPWP. Because this paragraph was discussing the choice of preindustrial dataset, we do not agree that the mPWP F&D dataset will be an issue here.

[TDH13]: However this is not born out with modern sediments in the region so I think this is special pleading! Large compilations, the most recent being Tierney and Tingley (2018) do not identify an upwelling bias in the modern data set. [line 540]

The sentence referred to in the comment is: "This lends some credence to the idea that the observed mismatch between PlioMIP2 $\Delta$SST and the F&D19_30 proxy-based anomaly could arise from the complexities/uncertainties associated with interpreting alkenone-based SSTs in the region as simply an indication of mean annual SST (Leduc et al. 2014)." The sentence was written as a possible explanation for the data-model mismatch. We also give other possible reasons for the data-model mismatch (i.e. that the model's struggle in upwelling regions). Although there is not a seasonal upwelling

bias in the modern dataset, it is not implausible that such a bias could exist in the Pliocene. In a revised version of the manuscript we will continue to include this as a possible reason for the Pliocene model-data mismatch but will also include that no such upwelling bias exists in the modern dataset in order to include additional information for the reader.

[TDH14]: My concern here would be how much the use of the F&D data set alters this new estimate in comparison to Hargreaves and Annan? [lines 555-565]

This point refers to how the estimate of ECS depends on the dataset chosen. We agree that the estimate of ECS depends on the dataset chosen, but note that it also depends on the relationship between the published ECS and the Pliocene warming in the models. We suggest that using the new F&D dataset, instead of the PRISM3 SST anomaly field used by Hargreaves and Annan would improve the estimate of ECS. This is because the PRISM3 SST anomaly field was based on warm peak averaging over a 240,000-year timeslab while the F&D dataset better represents the MIS KM5c timeslice that our models are set up to represent. However, we will note in the revised version of the manuscript that as more orbitally tuned SST data becomes available it will be important to revisit the ECS analysis to ensure maximum accuracy.

[TDH15]: The quality of the estimation also depends on the reliability of the paleo-$CO_2$ estimates

We agree that the relationship between ECS and ESS will only be robust if we are using the correct palaeo $CO_2$ value. Please see the response to TDH4, which explains why we are not able to quantify errors due to $CO_2$ uncertainties at the moment, but how we plan to do this in the future.

---

## Referee Report (RR1)

I read the author's responses to both my comments and that of another reviewer who has considerably more expertise in climate modeling than I do. The authors clearly evaluate the concerns raised and to the most part address them in this revision.

While the authors choose to revise their title, I find it awkward, specifically the construction of a descriptive component and an active one that is vague: "large scale climate features and constraining sensitivity". What kind of sensitivity is left unanswered, although I imagine most people will infer it as ESS to CO2.

That said, I find the current revision clearly superior to the initial submission and full of insightful information. I raised questions in my previous review as to the quality of the Foley/Dowsett proxy SST data set but on re-examination, and based on the author's response, I don't think this is a first-order limitation to their results.

SPECIFICS:

I am not sure if the response to my earlier query is satisfied:

[TDH6]: I would have thought that reduction in winter sea ice and/or lower land surface albedo would have generated a larger winter warming relative to the mean anomaly. [lines 230-240].
*The amount of winter sea ice has very little effect on temperature. This is because there is no sunlight over the winter pole, and so the value of the surface albedo is irrelevant.*
*The reduction in winter snow cover away from the pole will affect only be a small proportion of the northern hemisphere surface. Therefore, it can only have a limited effect on hemisphere averaged temperatures*

**My understanding is that sea ice has an important effect not through albedo, but by capping off the surface ocean which has a large capacity to buffer atmospheric cooling in the winter. My query comes from studies that suggest that extensive sea ice during the younger Dryas, for example, led to very pronounced winter temperature anomalies. Likewise, I based the land albedo comment on reading that winter anomalies are reduced when tundra is replaced by denser vegetation. Can the authors comment?**

**The revised manuscript has this sentence in the introduction:**

"Modelled sea-ice responses were studied by Howell et al. (2016), who demonstrated a significant decline in Artic sea-ice extent, with some models simulating a seasonally sea-ice free Arctic Ocean driving polar amplification of the warming. "

**I would note that the revised manuscript has more discussion of seasonality and the timing of seasonal temperature maxima (p 19-20) that are very interesting and useful to a proxy person.**

Likewise, I'm not sure the response to this comment hits the nail on the head

[TDH9]: To me, the pattern of data anomalies exceeding model anomalies near the gyre boundaries is quite robust (see MyClymont or example) and is a general feature of "warm climate" reconstructions- see Brierley for an earlier Pliocene time slice, and many others for the Miocene and Eocene. I think there is a fundamental model deficiency here. [line 401-406]
*We agree that there could be a fundamental model deficiency in these regions and have stated near line 520 "The simulation of upwelling systems is particularly challenging for global numerical climate models due to the spatial scale of the physical processes involved, and the capability of models to represent changes in the structure of the water column (thermocline depth) as well as cloud/surface temperature feedbacks". However, the interpretation of data in upwelling regions is also not trivial and we also discuss this in our paper.*

**The movement of SST gradients well poleward from the present gyre boundaries is a robust feature of warm climate SST reconstructions. Whether this actually represents extending the gyres is another question implied by the SST data but not proven. But I do not consider gyre boundaries to be "upwelling zones"- perhaps the authors misunderstood my comment. The gyre boundaries today represent major areas of thermocline ventilation and generally where mode waters form- very distinct processes from my understanding of "upwelling". I think the problem of the models is much larger than a failure to simulate upwelling , which of course depends a lot on model resolution, coastline resolution etc.**

**I would also like the authors to extend their evaluation of upwelling biases in alkenone SST: Benguela is not the only instance where there is no detectable upwelling bias. We wrote a paper on the California margin (Herbert et al., ) and one can also look at the Arabian Sea and Peru-Chile margin in vain for large upwelling-related anomalies.**

Herbert, T.D., J.D. Schuffert, D. Thomas, K. Lange, A. Weinheimer, and J.-C. Herguera, 1998, Depth and seasonality of alkenone production along the California margin inferred from a core-top transect, *Paleoceanography*, **13**: 263-271.

---

## Author Response (AR2)

We would like to thank the reviewers for reading our paper again and for providing additional comments. We have now addressed these comments. The comments are given below, and our response is in italics.

Reviewer 1

I am not sure if the response to my earlier query is satisfied:

[TDH6]: I would have thought that reduction in winter sea ice and/or lower land surface albedo would have generated a larger winter warming relative to the mean anomaly. [lines 230-240].

*The amount of winter sea ice has very little effect on temperature. This is because there is no sunlight over the winter pole, and so the value of the surface albedo is irrelevant.*
*The reduction in winter snow cover away from the pole will affect only be a small proportion of the northern hemisphere surface. Therefore, it can only have a limited effect on hemisphere averaged temperatures*

My understanding is that sea ice has an important effect not through albedo, but by capping off the surface ocean which has a large capacity to buffer atmospheric cooling in the winter. My query comes from studies that suggest that extensive sea ice during the younger Dryas, for example, led to very pronounced winter temperature anomalies. Likewise, I based the land albedo comment on reading that winter anomalies are reduced when tundra is replaced by denser vegetation. Can the authors comment?
The revised manuscript has this sentence in the introduction:
"Modelled sea-ice responses were studied by Howell et al. (2016), who demonstrated a significant decline in Artic sea-ice extent, with some models simulating a seasonally sea-ice free Arctic Ocean driving polar amplification of the warming. "
I would note that the revised manuscript has more discussion of seasonality and the timing of seasonal temperature maxima (p 19-20) that are very interesting and useful to a proxy person. Likewise, I'm not sure the response to this comment hits the nail on the head

*We agree that our response to this comment may have been oversimplified, and that sea ice can affect temperature in a multitude of ways. A discussion of PlioMIP2 sea ice is presented in Nooijer et al (this issue). However, looking at their paper there does not seem to be a large correlation between winter SAT warming and winter sea ice loss in PlioMIP2; those models which show largest winter warming do not necessarily show the greatest loss of winter sea ice. (However, there is a strong correlation between the two in the annual mean). Overall, it appears that the effect of winter sea ice loss on SAT is complicated and is beyond the scope of the current study.*

*We are pleased that the reviewer appreciated the additional discussion of seasonality that is in the revised version of the paper. We agree that it is an improvement over the previous version.*

[TDH9]: To me, the pattern of data anomalies exceeding model anomalies near the gyre boundaries is quite robust (see MyClymont or example) and is a general feature of "warm climate" reconstructions- see Brierley for an earlier Pliocene time slice, and many others for the Miocene and Eocene. I think there is a fundamental model deficiency here. [line 401-406]

*We agree that there could be a fundamental model deficiency in these regions and have stated near line 520 "The simulation of upwelling systems is particularly challenging for global numerical climate models due to the spatial scale of the physical processes involved, and the capability of models to represent changes in the structure of the water column (thermocline depth) as well as cloud/surface temperature feedbacks". However, the interpretation of data in upwelling regions is also not trivial and we also discuss this in our paper.*

The movement of SST gradients well poleward from the present gyre boundaries is a robust feature of warm climate SST reconstructions. Whether this actually represents extending the gyres is another question implied by the SST data but not proven. But I do not consider gyre boundaries to be "upwelling zones"- perhaps the authors misunderstood my comment. The gyre boundaries today represent major areas of thermocline ventilation and generally where mode waters form- very distinct processes from my understanding of "upwelling". I think the problem of the models is much larger than a failure to simulate upwelling , which of course depends a lot on model resolution, coastline resolution etc.

I would also like the authors to extend their evaluation of upwelling biases in alkenone SST: Benguela is not the only instance where there is no detectable upwelling bias. We wrote a paper on the California margin (Herbert et al., ) and one can also look at the Arabian Sea and Peru-Chile margin in vain for large upwelling-related anomalies.

Herbert, T.D., J.D. Schuffert, D. Thomas, K. Lange, A. Weinheimer, and J.-C. Herguera, 1998, Depth and seasonality of alkenone production along the California margin inferred from a core-top transect, Paleoceanography, 13: 263-271.

*The reviewer raises an interesting point about there being no detectable upwelling bias in many regions. However, for this paper we think it is sufficient to limit our discussion to the Benguela region. This is because it is the Benguela region that shows systematic model-data disagreement in PlioMIP2. We do not see model-data disagreement at the California margin, the Arabian Sea or the Peru-Chile margin.*

*We do not perform an analysis of the location of the gyres on the PlioMIP2 models in this paper, and therefore would not like to comment on how the models reproduce these features - this could be an interesting topic for a future paper. However, we now point out that the absolute Pliocene SSTs from the model is not warm enough in the position of the modern North Atlantic gyre.*

Reviewer 2

The revised version of the manuscript by Haywood et al. is improved over the original manuscript. I especially appreciate their inclusion of closer comparisons with PlioMIP1. As I said in my original review, as an analysis of a model intercomparison project, the manuscript is fine and some aspects of the study are especially interesting. In my original review, I suggested removing the climate and earth sensitivity analysis. The authors have decided not to do that. I'm okay with that decision. Personally, I find the manuscript to be too long and meandering, and that the climate sensitivity results would be more impactful if presented on their own.

The presentation of the manuscript is uneven. I still object to the introduction (lines 61-84), which remains a list of prior publications, and that the introduction does not explain the point of the paper. I suggested a justification in my previous review, which the authors chose to ignore. I don't think the fact that the MP has been studied for more than 25 years (lines 53-55) is a good justification for continuing to do so. The end of the Discussion (lines 670-675) provides a more compelling reason, that the Pliocene may have some lessons for near future climate change. I would encourage the authors to consider introducing these ideas in the Introduction.

*Since this is the first paper presenting the main features of PlioMIP2 the introduction aimed to provide:*

- a) *A full recap of what had been achieved in PlioMIP1*
- b) *Specific differences between PlioMIP1 and PlioMIP2 and why these differences might be important*

*We believe that the introduction accomplished our objectives as defined above.*

*However, we accept the reviewer's comment that we could have provided more information about the importance of studying the Pliocene climate.  We therefore rewrite the first paragraph of the introduction as:*

*"Efforts to understand climate dynamics during the mid-Piacenzian Warm Period (MP; 3.264 to 3.025 million years ago), previously referred to as the mid-Pliocene Warm Period, have been ongoing for more than 25 years. This is because the study of the mPWP enables us to address important scientific questions. The inclusion of a Pliocene experiment within the CMIP6 experimental protocols underlines the general potential of the Pliocene to address questions regarding the long-term sensitivity of climate, and environments, to forcing as well as the determination of Climate Sensitivity specifically."*

Below I list some specific comments:

L. 136-137. "The standard version of the PRISM4 boundary conditions provides the best possible…" If these are the "best possible" set of boundary conditions, then why is there an "enhanced" set? (This is confusing.) It's also worth stating in the manuscript that all but one model uses the "enhanced" boundary conditions.

*Originally we wrote "The standard version of the PRISM4 boundary conditions provides the best possible realisation of Pliocene conditions **based around a modern land/sea mask**. The enhanced boundary conditions include all reconstructed changes to the land/sea mask and ocean bathymetry." This was intended to imply that the enhanced was better because it required land sea mask changes, however the standard was the best possible if land sea mask changes could not be carried out.*

*For clarity we have reordered the text around this sentence and now write.*

*"Two versions of the PRISM4 boundary conditions were produced known as enhanced and standard. The enhanced version comprises all PRISM4 boundary conditions including all reconstructed changes to the land/sea mask and ocean bathymetry.  However, groups which are unable to change their land/sea mask can use the standard version of the PRISM4 boundary conditions, which provides the best possible realisation of Pliocene conditions based around a modern land/sea mask. In practice all models except MRI-CGCM2.3 were able to utilise the enhanced boundary conditions."*

L. 162. "minimum of 500 simulated years" It might be worth briefly noting that all but two models were run for 1000 yrs or more. Some of the models are quite far from TOA radiation balance. For this reason, the authors should add the $\Delta$SAT for the last 100 years of each model in Supplemental Fig. 1.

*We now write "Integration length was set to be 'as long as possible', or a minimum of 500 simulated years, however all but two of the modelling groups in PlioMIP2 contributed simulations that were in excess of 1000 years (supplementary table 1).*

*As suggested, we also add the drift in the Plio_Core – Pi_Ctl anomaly to supplementary table 1.  For your information, a full table showing the drift in Plio_Core, Pi_Ctl and the Plio_Core-Pi_Ctl anomaly is below, it is clear that the drift implies that all simulations are sufficiently spun-up.*

| Model name | Drift plio_core (degC/ centuary) | Drift pi_ctl (degC/centuary) | Drift (plio_core – pi_ctl) (degC/centuary) |
|---|---|---|---|
| CCSM4 | -0.11 | -0.06 | -0.05 |
| CCSM4-UoT | 0.07 | 0.03 | 0.04 |
| CCSM4-Utr | -0.07 | -0.1 | 0.02 |
| CESM1.2 | -0.22 | -0.15 | -0.07 |
| CESM2 | -0.14 | -0.03 | -0.1 |
| COSMOS | -0.05 | 0.5 | -0.1 |
| EC-Earth3.3 | 0.07 | 0.1 | -0.03 |
| GISS2.1G | 0.06 | -0.13 | 0.18 |
| HadCM3 | 0.19 | 0.11 | 0.08 |
| IPSLCM5A | 0.02 | 0.17 | -0.15 |
| IPSLCM5A2 | -0.08 | 0.01 | -0.09 |

| IPSLCM6A | 0.1 | -0.3 | 0.4 |
| MIROC4m | 0.03 | -0.06 | 0.08 |
| MRI2.3 | 0.08 | -0.1 | 0.18 |
| NorESM1-F | -0.17 | -0.21 | 0.04 |
| NorESM-L | -0.21 | -0.08 | -0.13 |

L. 186. This estimate of ESS assumes that the sensitivity to CO2 is linear. We know this is not necessarily the case (e.g. Caballero and Huber, PNAS, 2013; Zhu et al., Science Advances, 2019). The authors should discuss this in their list of errors in the estimate of ESS (line 189).

*The text has been changed near line 190 to account for the nonlinear relationship between CO2 and ESS*

L. 212. "In general, PlioMIP1 models…" This sentence is confusing. I don't think (though I'm not sure because it doesn't make sense to me) that the sentence is correct as written.

*Since this sentence didn't make sense, we have removed it, as it wasn't necessary to the paper.*

L. 216-217. "The larger ensemble mean…rather than an increase in the temperature anomaly due to the change in boundary conditions." This is an example of some of the clumsy writing that pops up throughout the manuscript. An ensemble mean temperature cannot increase by adding models that have the same temperature response as the mean. The authors intend to write that "The increase in the ensemble mean in PlioMIP2 is due to the addition of models with a large temperature sensitivity to the boundary conditions, rather than an increase in the temperature response of the former PlioMIP1 models to changes in boundary conditions."

*We now write: "The ensemble mean temperature anomaly is larger in PlioMIP2 than in PlioMIP1 because of the addition of new and more sensitive models to PlioMIP2, rather than being due to the change in boundary conditions between PlioMIP1 and PlioMIP2. "*

L. 378. "Some models show a different seasonal cycle to the annual mean.." I can't make sense of why there is a comparison of the seasonal cycle with the annual mean. Perhaps the authors simply mean that the seasonal cycle varies between models.

*Many apologies. This was a typographical error. We compared the seasonal cycle in each model with **the ensemble** mean, not the **annual** mean. This has now been corrected.*

L. 407. "is representing a state in which the associated feedbacks are in equilibrium." How do the authors know this, and what is the point? Is there evidence that the climate is in equilibrium at any point in time?

*The sentence this point refers to is: "Due to the prescribed changes to ice sheets and vegetation, the Plio$_{Core}$ simulation is representing a state in which the associated feedbacks are in equilibrium."*

*To improve clarity we now write: "Since ice sheet changes were prescribed, there will be no transient response due to ice sheet changes and the PlioCore experiment will be in equilibrium with the ice sheets."". (Note: we remove the reference to vegetation because one of the models – COSMOS – was run with dynamic vegetation).*

L. 537. "A particularly robust feature…over the modern Sahara Desert…" Indicate what the feature is, i.e. "A particularly robust feature across the ensemble is an increase in precipitation over the Saharan Desert and Asian monsoon regions…"

*The original sentence was:*

*"A particularly robust feature of precipitation change across the ensemble is over the modern Sahara Desert and over the Asian monsoon region (Figure 5d)."*

*We change the sentence as suggested by the reviewer.*

L. 549-551. I'm not sure what the point is here. Given that the present atmospheric CO2 level now exceeds that of the mid-Pliocene, I think this is a questionable statement. Anyway, it's the differences between the mid-Pliocene and the modern that are interesting and scientifically important.

*The sentence this point refers to is: "Nonetheless the similarities between the general features of the Pliocene experiments and future experiments continues to support the use of the Pliocene as one of the best geological analogues for the near future (Burke et al. 2018), despite the different boundary conditions."*

*We do not think this statement is as questionable as the reviewer makes out. We do not state that it is an ideal geological analogue, rather that it is one of the best paleoclimate analogues and have compared the PlioMIP2 results to near future predictions throughout the manuscript, showing why this is the case.*

L. 556. The use of "release date" as a model characteristic is misguided and (forgive me) lazy. It implies that newer is better, and it is a shortcut that avoids a discussion of the changes in model parameterizations that are responsible for the change in sensitivity. As in my first review, I encourage the authors not to use release date in the manuscript and instead to briefly summarize some of the reasons for changes in sensitivity that have been recently published in the literature.

*Following this comment, we have decided to remove all places where release date was used as a model characteristic from the paper. However, we still include our discussion as to how climate sensitivity changes as models are developed within a family.*

[revised manuscript text omitted]